# Processes governing snow ablation in alpine terrain. Detailed measurements from the Canadian Rockies

**Michael Schirmer[1,2] and John W. Pomeroy[1]**

[1] Centre for Hydrology, University of Saskatchewan, Canada

[2] WSL Institute for Snow and Avalanche Research SLF, Davos, Switzerland

Correspondence to: Michael Schirmer (michael.schirmer@slf.ch)

**Abstract**

The spatial distribution of snow water equivalent (SWE) and melt are important to estimating areal melt rates and snowcover depletion dynamics but are rarely measured in detail during the late ablation period. This study contributes results from high resolution observations made using large numbers of sequential aerial photographs taken from an Unmanned Aerial Vehicle on an alpine ridge in the Fortress Mountain Snow Laboratory in the Canadian Rocky Mountains from May to July. With Structure-from-Motion and thresholding techniques, spatial maps of snow depth, snowcover and differences in snow depth (dHS) during ablation were generated in very high resolution as proxies for spatial SWE, spatial ablation rates, and snowcover depletion (SCD). The results indicate that the initial distribution of snow depth was highly variable due to overwinter snow redistribution and the subsequent distribution of dHS was also variable due to albedo, slope/aspect and other unaccountable differences. However, the initial distribution of snow depth was five times more variable than that of subsequent dHS values which varied by a factor of two between north and south aspects. dHS patterns were somewhat spatially persistent over time but had an insubstantial impact on SCD curves, which were overwhelmingly governed by the initial distribution of snow depth. The reasons for this are that only a weak spatial correlation developed between initial snow depth and dHS. Previous research has shown that spatial correlations between SWE and ablation rates can strongly influence SCD curves. Reasons for a missing correlation in this study area were analysed and a generalisation to other regions were discussed: What is needed for a large spatial correlation between initial snow depth and dHS and when should

they taken into account to correctly modelling SCD. These findings suggest that hydrological and atmospheric models need to incorporate realistic distributions of SWE, melt energy and cold content and so must account for realistic correlations (i.e. not too large or too small) between SWE and melt in order to accurately model snowcover depletion.

## 1    Introduction

The spatial variability of snow water equivalent (SWE) exerts an important control on catchment or grid-averaged snowmelt (Pomeroy et al., 1998; Liston, 1999). When focussing on complex terrain with only minor vegetation effects on SWE distribution, differences in precipitation, snow redistribution, melt energy and freezing levels lead to a spatially variable distribution of SWE (e.g. Clark et al., 2011). For modellers of snow-hydrological applications the question arises as to which of those processes need to be considered. It is well known that south-facing slopes receive more melt energy than do north-facing slopes due to differences in solar radiation. At 50°N on April 1, the differences are already 40% for a slightly inclined slope of 10°, however, these differences decrease as summer solstice approaches (Figure 1, Gray and Male, 1981). It is also well known that SWE distribution at peak accumulation is highly variable in alpine terrain. Liston et al. (2004) presented maps with regional differences of coefficients of variation (CV) of snow depth. For alpine regions a CV of 0.85 is suggested. Both, the variability in melt energy and SWE influence snow cover depletion. This can be visualized in snow-cover depletion curves, which are a function of snow-covered area (SCA) over time or grid-averaged SWE (e.g. Essery and Pomeroy, 2004; Clark et al., 2011). Both studies illustrate with theoretical simulations how increasing melt rate and peak SWE variability change the rate of areal snow-cover depletion. From their theoretical illustrations (Fig. 3 and 4 in Clark et al., 2011), it is clear that in alpine regions with a large variability in melt rate and peak SWE, ignoring SWE rather than melt rate variability would be the greater modelling mistake. However, as Pomeroy et al. (2004) pointed out, the importance of melt variability on SCD increases if a spatial correlation between melt and SWE exists. This suggests that in alpine terrain the question of relative contribution of spatially variable melt rates or snow redistribution on SCD can be reduced to the question of whether such a correlation between melt and SWE exists and how large it is.

Besides theoretical considerations, there are a number of existing field and modelling studies on the relative importance of spatially variable melt or snow redistribution on SCD.

There are studies that have found the temporal progression of snow-cover depletion (SCD) to be governed primarily by the variability caused by snow redistribution rather than the variability caused by melt rate differences (Shook and Gray, 1994; Luce et al., 1998; Anderton, 2004; Egli et al., 2012). These studies show with spatial observations that snow cover depletion (SCD) can be modelled with statistics derived during peak accumulation. Luce et al. (1998) modelled snow cover depletion with a spatially distributed energy balance model which integrated drifting snow redistribution based on an empirically derived drift factor. Ignoring this drift factor deteriorated model results, which suggests the relative importance of snow redistribution over melt variability. Grünewald et al. (2010) made indirect measurements of spatial melt rate and SWE via snow depth (HS) and the change in HS (dHS) by terrestrial LiDAR and applying a few measured bulk densities to estimate SWE and ablation rates. They found that SWE and melt rate were spatially uncorrelated over most of their ablation season, except for a correlation coefficient of r = -0.4 for one sub period. They noted that the variability of SWE was much larger than the variability of melt rates. In the same study area over an additional winter season Egli et al. (2012) calculated SCD curves that assumed correlations between HS and the change in HS (dHS), however these curves deviated substantially from observations, suggesting that such correlations did not exist. Neither study examined why such correlations were absent.

On the other side, spatially varying melt rates – caused by differences in insolation due to aspect (Marks and Dozier, 1992), net solar irradiance due to albedo differences (Skiles et al., 2015), internal energy storage due to deep, cold snow (DeBeer and Pomeroy, 2010) and advected energy due to bare ground (Mott et al., 2011; 2013) can alter this pre-melt SWE distribution and, when correlated to SWE, result in substantial changes to SCD curves (Pomeroy et al., 2004; Essery and Pomeroy, 2004; DeBeer and Pomeroy, 2010). Pomeroy et al. (2003) took measurements of energy fluxes to snowpacks using eddy correlation and slope based radiometers and snow ablation using spatially distributed snow surveys in a Yukon mountain valley in April and found that whilst ablation was proceeding rapidly on south facing slopes where snow was initially shallow, snow accumulation was still occurring on north facing slopes where a large drift had formed. In Yukon and the Canadian Rockies, subsequent studies found melt variations to be important in controlling snow ablation and SCD (Pomeroy et al., 2003; 2004; Dornes et al., 2008a, b; DeBeer and Pomeroy, 2009; 2010). Pomeroy et al. (2004) reported that different spatial scales and landscape classes influence melt rates to be positively or negatively correlated to pre-melt SWE throughout melt in a wide

variety of cold regions mountain environments. DeBeer and Pomeroy (2010) found in a windswept alpine catchment subject to substantial snow redistribution, that melt rate variations were important in the Canadian Rockies during early melt. Winstral and Marks (2014) found modelled SWE and melt rates were correlated with r = -0.66 in the mountains of southern Idaho. Such a large correlation between modelled melt and SWE would indicate that spatial melt differences are relevant to model SCD correctly in some regions. Dornes et al. (2008a, b) found that hydrological models and land surface schemes that did not consider slope and aspect impacts on melt as well as initial SWE could not be calibrated to produce realistic SCD curves or streamflow discharge hydrographs. Furthermore, they showed substantial interannual variability of SWE distributions and correlations to melt rate. Interannual differences on the interplay of SWE and melt variability will enhance the problem of correctly modelling SCD.

Regional differences, e.g. the complexity of the terrain and wind redistribution, will alter the dominance of SWE variability on SCD and may thus explain part of the different findings of various studies. Not all cited studies fall in the highest CV category suggested by Liston et al. (2004). Furthermore, in study regions with a large elevation gradients, altitudinal melt energy differences as well as precipitation phase differences will play an important role in governing SCD (Blöschl and Kirnbauer, 1992; Elder et al., 1998).

From a practical modelling perspective, it is simpler to explicitly calculate melt energy differences in a model (Marks and Dozier, 1992) than to calculate snow redistribution mechanistically over complex terrain (Liston and Sturm, 1998; Mott et al., 2010; Fang et al., 2013; Musselman et al., 2015). Empirical modelling of SWE variability (Luce et al., 1999; Winstral and Marks, 2002; Liston et al., 2004, Essery and Pomeroy, 2004; Helbig et al., 2015) has therefore been a preferred choice. As Essery and Pomeroy (2004) pointed out it is relevant to consider also the spatial correlation between melt and SWE to correctly model SCD. One attempt in this direction is shown in DeBeer and Pomeroy (2010) who modelled this correlation based on the cold content of snow. High-resolution gridded model approaches with energy balance models include this effect as well, but only with a realistic SWE distribution one can assume also a realistic correlation to spatially modelled melt rates. Brauchli et al. (2017) extended the method of Luce et al. (1998) to higher spatial resolution, i.e. scaling measured solid precipitation to observations using LiDAR data. Without discussing the effect of an improved correlation between modelled melt and SWE compared

to standard methods of distributing precipitation, Brauchli et al. (2017) were able to improve

modelled runoff for certain periods.

This study wants to assess the the influence of peak SWE variability, melt rate

variability and especially their spatial correlations on SCD in alpine terrain using high-

resolution distributed measurements rather than sparse manual sampling or relying on model

results. The use of high resolution measurements is potentially important because peak alpine

snow depth, and thus also peak SWE, is known to vary most substantially below the scale of

tens of metres in alpine environments. The coarse-resolution manual probing of previous

studies may have missed important spatial structures which may determine the results (Clark

et al., 2011). Most studies on this topic have relied on modelled melt rates, even though there

are substantial uncertainties in melt modelling over complex terrain.  For instance, Mott et al.

(2011) were only partly successful in high resolution modelling of alpine melt rate variations.

Those studies which have used high-resolution distributed snow depths, such as Egli et al.

(2012) did not attempt to diagnose the variation of and correlation between SWE and melt

rates.

**2  Data and methods**

**2.1  Site description**

A study region was chosen which showed substantial differences in aspect and slope

to ensure spatial melt differences. In a nearby study site DeBeer and Pomeroy (2010) found

spatial melt rates to be important for snowcover depletion, at least during early melt. Large

drifts commonly form on south facing slopes in this area (MacDonald et al., 2010; Musselman

et al., 2015) suggesting a correlation between melt energy and SWE. The study area is located

in the Canadian Rocky Mountains in southern Alberta, Canada. Figure 2a shows a

topographic map of the study area, an alpine ridge in a NE – SW orientation. Gullies and

small scale aspect variations can be found in the slopes on both sides of the ridge. Extreme

south and north aspects are underrepresented in the terrain snowcovered at the beginning of

the study period (Fig. 2b). The snowcovered area is reasonably steep with two peaks in the

slope distribution at $10^o$ and $25^o$ (Fig. 2c). On both sides of the ridge the slope steepens to up

to $40^o$. Vegetation played a role in snow deposition patterns, mainly in the lee of shrubs and

clusters of small trees in krummholz with heights up to 2 m. Areas within these vegetation

clusters were excluded from the study as vegetation degraded the digital surface models (DSMs) derived from UAV SfM photogrammetry (see section 2.4). The included area only covered bare or sparsely vegetated ground so that vegetation effects can be excluded.

Two weather stations are located at the ridge, one on top of the ridge (Fortress Ridge - FRG) and one in a south facing slope (Fortress Ridge South - FRS, Fig. 2a). The local Fortress Mountain Snow Laboratory within the regional Canadian Rockies Hydrological Observatory provided five more weather stations within less than 2 km distance of the ridge, which were used to interpret weather situations and for quality control.

## 2.2   Unmanned aerial vehicle (UAV) Data acquisition

A "Sensefly Ebee RTK" fixed wing UAV was used with a modified consumer-grade Canon Elph compact RGB camera. As a base station a Leica GS15 differential GPS system communicated with the UAV to tag captured images with corrected geolocations. Additionally, ground control points were measured with this differential GPS system, which improved the quality of the Digital Surface Models (DSMs) generated. For a more detailed description of the UAV and the usage please refer to Harder et al. (2016).  An area of 0.31 km$^2$ (666 x 470 m) was separated into two subareas because of battery restrictions on flight coverage (Fig. 2a, red polygons). Each flight lasted approximately 20 min. The flight altitude was 100 m over the ridgetop, which resulted in an approximate resolution (ground sampling distance) of less than 4 cm.  A lateral overlap of the images of 85% and a longitudinal overlap of 75% was chosen as suggested by the manufacturer for difficult terrain. Ideally, four flights in total were made each sampling day, two for each subarea with perpendicular flight plans, which is suggested by the manufacturer for complex terrain. Weather conditions and technical problems often allowed only a part of this program. Wind speeds over 14 m/s or occurrence of precipitation restricted flying, while camera malfunctions or connection issues with the Leica GPS base station were the most typical technical limitations. In total, the UAV was flown over snow from 15 May to 24 June 2015 at eight different days with substantial depth differences and four flights over bare ground on 24 July 2015. However, as stated section 3.2, we had to restrict analysis to two melt periods.

## 2.3 Accuracy evaluation and manual measurements

The accuracy assessment of this rather new method to determine snow depth was given a high priority and is described in detail by Harder et al. (2016) for this environment and others. In short, 100 differential GPS measurements on bare ground were taken. Approximately 60% of the area was bare at the beginning of the study period which allowed distribution of GPS ground measurements over a large part of the study area (Fig. 3a) and thus widespread detection of any general misalignment of DSMs or local tilts. These points could be used for all available flights. Differential GPS measurements were also taken at the snow surface on the day of the specific flights, but technical problems often allowed only limited additional time for these surveys. For most of the days up to 20 differential GPS measurements on snow could be taken. At these GPS measurement points, snow depth was also manually measured and snow density was measured at approximately each third of these points. Density measurements were not sufficient to confidently estimate SWE from snow depth into SWE and ablation rates from differences in snow depth. As such HS and dHS are analysed and interpreted as proxies for SWE and melt in the text. Since redistribution and compaction of snow were not a relevant process for this time period late in the ablation season, dHS can serve as a reasonable proxy for melt.

Harder et al. (2016) described errors and accuracies of the UAV measurements in detail. In short, from 100 measurements on bare ground, the root mean square errors of bare ground surface elevation ranged between 4 and 15 cm with a mean of less than 9 cm. Over snow with fewer measurements an increase in these error measures could not be detected. A signal-to-noise ratio (SNR) was used to ensure that the signal of the UAV was sufficiently larger than the error, defined as mean of the signal divided by the standard deviation of the error. The potential impact of this error on the results presented is discussed in section 3.2.

## 2.4 Spatial data generation

Digital Surface Models (DSMs) and orthomosaics were created by application of SfM techniques (Westoby et al., 2012) using the software Postflight Terra 3D, which was provided with the UAV. Default settings likely resulted in overexposed pixels, which created erroneous points in the point cloud over snow that appeared several metres above the real snow surface. This issue could partly be solved with a semi-global-matching option within the software, which reduced the number of affected areas. Remaining areas with errors were manually

excluded (Harder et al., 2016). DSMs and orthomosaics were resampled to a common grid and resolution of 10 cm, which increased the speed of subsequent data analysis substantially.

Subtracting DSMs provided both snow depth (HS) and differences in snow depth (dHS). dHS was scaled by the time interval between observations for comparison of varying observation periods. Snow-covered area (SCA) was defined using individual thresholds in RGB values for different flights using the orthomosaics. Manual adjustment was needed to ensure that very dark snow was classified correctly (see for example Fig. 3b). HS was masked by the SCA of the date of the flight, whilst dHS was masked by the SCA on the dates of the first and subsequent flight. Figure 3c and 3d show examples of HS and dHS maps of a part of the study area. Several areas such as ski lifts and snow cat tracks and erroneous points as mentioned above were excluded from the analysis. Furthermore, large errors were detected in areas close to vegetation, which were manually excluded. The marked green area in Fig. 3d indicates the excluded area for this part of the study area.

To explain the observed differences in snow depth, several topographical variables were created using the DSMs. *Deviation from North* and *Slope* were calculated on a 1 m resolution to exclude small-scale noise of the DSMs. *Solar Irradiance* was calculated for a 1 m resolution for each flight day with the Area Solar Radiation function in ArcGIS. To account for albedo differences the *Brightness* of the orthomosiac pixels was abstracted on a 10 cm resolution. The blue value was chosen since it was least affected by unwanted illumination differences due aspect variations. *Brightness* and *Solar Irradiance* are temporal averages based on the first and the last flight and, if available, flights within the periods.

## 2.5  Data analysis

To diagnose reasons for spatial differences in snow depth change (dHS), Pearson correlation coefficients were calculated with several potential explaining variables as *Slope*, *Deviation from North*, *Solar Irradiance*, *Brightness*, current snow depth (HS) and snow depth at the beginning of the study period (HS0). Scatter plots also were visually inspected to detect reasons for strong or weak correlations or non-linear dependencies. The scatter plots were too dense to interpret visually because of the high resolution and so instead of plotting point pairs, the density of point pairs in a limited area of the plot was visualized (e.g. in Fig. 5a).

Spatial dependencies of the spatial structure of dHS and its correlation with explaining variables were analysed with variograms and correlograms. Variograms were calculated with

$$\hat{\gamma}_x(h) = \frac{1}{2|N(h)|} \sum_{(i,j) \in N(h)} (x_j - x_i)^2, \qquad (1)$$

2      for point pairs $x_i$ and $x_j$ in a lag distance class $N(h)$ (e.g. Webster and Oliver, 2007).

3 Correlations between two variables $x$ and $y$ in a certain lag distance $h$ were calculated with the

4 cross-variogram as an estimator of the covariance (Webster and Oliver, 2007).

$$\hat{\gamma}_{xy}(h) = \frac{1}{2|N(h)|} \sum_{(i,j) \in N(h)} (x_j - x_i)(y_j - y_i), \qquad (2)$$

6      This covariance was scaled with estimators of the variance $\hat{\gamma}_x$ (Eq. 1) using

$$\rho_{xy}(h) = \frac{\hat{\gamma}_{xy}(h)^2}{\hat{\gamma}_x(h)\hat{\gamma}_y(h)}, \qquad (3)$$

to obtain a correlation measure (Webster and Oliver, 2007). Variograms and

correlograms were calculated only with a random subset of 10% of available data points to

save computational resources. Smallest number observations were $N > 5 \times 10^4$, which was

large enough to obtain consistent variograms and correlograms with different randomly

chosen subsets.

**3   Results and discussion**

**3.1   Overview and meteorology**

Fortress Ridge is well exposed to the wind, with peak hourly wind speeds over 20 m/s

and a mean of 4.6 m/s over the winter 2014/15 at the FRG station. Two dominant wind

directions can be identified, WSW and ESE, the latter is approximately perpendicular to the

ridge. The wind direction parallel to the ridge is associated with the highest wind speeds.

During precipitation and high wind events both directions were frequent. During late melt in

2015, wind speeds were substantially lower with a higher frequency of very calm days,

providing more frequent flying conditions for the UAV.

Due to high wind speeds, large parts of the ridge were snow-free during most of the

exceptionally warm and dry winter season. After a large late November 2014 snow storm, the

FRG station rarely documented snow on the ground and shallow snowpacks that did form

were regularly eroded by wind within a few days. The snow covered area (SCA) reached the seasonal maximum in late November after this substantial snowfall (80 mm) with light winds and dropped dramatically due to subsequent wind redistribution from blowing snow storms. When aerial measurements began on 19 May 2015, SCA was slightly below its typical winter value as spring ablation was under way. Without excluding any areas (see section 2.4) SCA was approximately 0.45 in both subareas (Fig. 3a).

Dust-on-snow was an obvious feature in late winter and the beginning of the melt season (Fig. 3b). It had not been observed to such extent in over a decade of observations in the region. This dust was locally eroded from the fine frost-shattered and saltation-pulverized shale particles at the ridge-top and was transported by wind to adjacent lee slopes and into gullies, similarly to wind-transported snow. Hence dust was deposited preferentially to snow drifts. Subsequent snow accumulation and melt processes led to a dust-on-snow pattern of high small-scale variability. The lower albedo from dust deposition may have influenced snowmelt energetics, but its high variability is different from the large scale, areally uniform dust deposition reported by Painter et al. (2010) where the dust source is in upwind arid zones and very fine aerosols are evenly deposited on snow.

Blowing snow transport and redistribution during the high wind speeds also caused a highly variable snow depth (Fig. 3c) as is expected in the region (Fang et al., 2013; Pomeroy et al., 2016). Snow was redistributed to the SE facing slopes of the ridge and also in gullies on the NW facing slopes, which are perpendicular to the ridge. Areas of bare ground and very deep snow (> 4 m) were only separated by a few metres distance. This high variability of snow depth at scales of from a few to tens of metres is a typical feature for wind-swept alpine snow covers (e.g. Pomeroy and Gray, 1995, p.22-27; Deems et al., 2006; Trujillo et al., 2007; Schirmer et al., 2011; Schirmer and Lehning, 2011). There is no avalanching redistribution of snow in the study domain.

An example of reductions in snow depth (dHS) due to ablation over a period of 13 days is shown in Fig. 3d. At the first glance differences between aspects are obvious, as well as smaller scale impact of albedo variations (cf. Fig. 3b). The driving forces to differences in ablation inferred from the observed differences in depth change will be examined in section 3.3.

The study covered the late melt period, when the highest ablation rates occurred. Peak SWE of 500 mm was measured with a weighing snow lysimeter (Sommer "Snow Scale") in a

nearby forest clearing on 20 April 2015. By the start of the study period on 19 May, SWE had gradually decreased to 300 mm, often interrupted by snowfall. During the study period after 19 May no significant (>3 cm) snowfall was observed. The much higher ablation rates compared to the previous weeks caused the snow to disappear at this station on 30 May. A very similar development could be observed at two other stations using snow depth sensors within the Fortress Mountain Snow Observatory, including the FRS station (c.f. Fig. 2a). On 30 May a SCA of 0.2 was measured from the UAV over the whole flight domain. Considering a typical pre-melt SCA of approximately 0.45, the presence of a significant SCA illustrates the value of spatially distributed measurements of snow ablation and cover, when all seven meteorological stations in the ~3 km$^2$ region were snow-free.

A meteorological overview during the study period is given in Fig. 4 at the FRG station (cf. Fig. 2a). Measurements of incoming shortwave radiation and air temperatures are shown on the left, and resulting modelled results with CRHM using Snobal as the melt module (cf. Fang et al., 2013) for a flat field simulation on the right. Although the FRG station was snow-free, CRHM was initialized with a hypothetical SWE amount of 800 mm in order to represent deeper nearby snow patches. Energy fluxes were summed and scaled for comparison over the indicated dates with UAV flights. The energy balance was dominated by inputs of net shortwave radiation. Modelled melt accelerated around 8 June when high incoming shortwave radiation was accompanied by smaller longwave radiation losses and larger sensible heat fluxes driven by air temperatures often in excess of 10$^o$ C.

### 3.2   Selection of melt periods

Melt periods were chosen to include sufficient ablation such that the dHS signal exceeded the measurement error from the UAV and data processing. A signal-to-noise ratio (SNR) was used, which relates the mean dHS with the typical standard deviation error (SD) found by Harder et al. (2016) for surfaces measured with the UAV to be 6.2 cm. Since two surface measurements are needed to achieve a dHS map, this SD value was doubled. For SNR ≥ 4, the signal is assumed to be sufficiently large to avoid mistaking it for a fluctuation in noise (Rose, 1973). Applying this criterion, mean dHS had to be larger than ~0.5 m. Given the availability of suitable flights in both subregions, this permitted two time periods for analysis; P1 from 19 May to 01 June 2015, and P2 from 01 June to 24 June 2015.

## 3.3  Factors influencing spatial differences in dHS

Table 1 shows the Pearson's correlation coefficient for above mentioned melt periods and different subareas. This univariate analysis shows clearly two driving factors for the earlier melt period, P1, albedo and solar radiation differences, expressed respectively with *Brightness* and either *Deviation from North* or with *Solar Irradiance*. The sign of the correlations is mainly as expected: More southerly and darker pixels showed larger dHS values. Exceptions (e.g. during P2 in the southern subarea) may be explained with observable differences between a few remaining snow patches with different albedo values, slope, snow depths and sky view factors. Energy contributions from longwave radiation (DeBeer and Pomeroy, 2009) or altered turbulent heat fluxes because of cold air pooling (Mott et al., 2011; 2016) may override an obvious relationship with solar radiation. Also, faster settling rather than melt of deeper snow is possible, although the snowpack was quite ripe.

In the first period, P1, *Brightness* had a large effect in the northern subarea (r = -0.66). Figure 5a visualizes this relationship between dark snow and dHS. The high scatter especially for brighter snow pixels can partly be explained with radiation differences. For the same period and area *Solar Irradiance* and *Deviation from North* had a correlation of r = 0.57. Figure 5b illustrates the dependency with *Solar Irradiance* but for white pixels only (approximately 50% of the observations). A clear dependency is visible with a correlation coefficient of r = 0.66. Radiation effects were more substantial during P2 in this northern subarea with r = 0.84 for both *Solar Irradiance* and *Deviation from North.* This may be explained due to less scatter produced by albedo differences in this period (r = 0.03). Darker parts of the snowcover melted out by the end of this period.

The correlations of dHS with *Brightness*, *Deviation from North* and *Solar Irradiance* were often strong. dHS increased from 5 to 7 cm/d (nearly 60% increase) as aspect shifted about 115 deg from north to south or snow from clean to dusty (c.f. Fig. 5b). This shows the importance of spatial variation in net solar irradiance to melt energetics – as exemplified by the modelled energy budget shown in Fig. 4b. The impact of dust on albedo and slope on solar irradiance is well established in the snow literature and so this is expected.

What is a more interesting finding here is that dHS was not correlated with initial HS0, Fig. 5c, Table 1), as was observed in other cold regions mountain studies in Canada such as DeBeer and Pomeroy (2009, 2010), Pomeroy et al. (2003, 2004), and Dornes et al. (2008 a, b). A lack of covariance between HS0 and dHS in late melt has important implications for

SCD curves (Pomeroy et al., 2001), which will be highlighted in section 3.5. Figure 5c shows the areal mean values for HS0 and dHS for flat areas (slope $< 5^{o}$) and areas on both sides of the ridge (threshold aspect is $235^{o}$, slope $\geq 5^{o}$). The hypothesis for this study period was that large drifts on south-facing parts of the ridge cause a correlation between melt energy and SWE. Indeed, the southeast part showed larger HS0 and dHS compared to the flat and northwest part of the study area. This suggests a correlation between HS0 and dHS, which was not apparent when analysing all pixels. In each subarea the range of snow depth was large, which diminished the observed correlation. More importantly, on the south-eastern face a mild negative correlation of r = -0.35 developed (Fig. 5d), which may be explained by a remaining cold content in deep drifts. This negative correlation is not apparent for smaller dHS values, in the northwest part of the ridge (Fig. 5e). The lack of correlation in the Fig. 5c point cloud can be interpreted as a compensation between the positive correlation driven by melt energy and the negative correlation from a cold content.

To aid in analysing the reasons for the lack of correlations between Hs and dHS in this study area one can formulate some prerequisites for large spatial correlations in general. For instance, cold content has the potential to establish a negative correlation since deeper snowpacks take longer to warm up to 0 $^{o}$C and so shallower snowpacks start melting earlier. This results in greater melt for shallower snowpacks. The spatial distribution of SWE and melt energy on slopes may result in a negative or positive correlations, which depend on whether deep drifts are found on north-facing or south-facing slopes. For a large correlation between Hs and dHS, either snow redistribution to slopes or deep snow cold content processes needs to be present and need to not counteract each other. In such a case the sign of the correlation driven by spatial distribution of SWE melt energy must be negative (drifts on north-facing slopes) and hence similar to the negative correlation driven by greater cold content in deeper snow. Remote sensing techniques can determine where deep drifts occur on north-facing slopes (Wayand et al., 2018; Painter et al., 2016) and these are quite prevalent in many regions. DeBeer and Pomeroy (2010) showed that spatial variation in cold content was large only in early melt and was unimportant to SCD later in the melt season when isothermal snowpacks predominate.

Given these scenarios some guidelines for modelling areal SCD can be provided. Models must be able to represent realistic correlations between SWE and melt in order to model the effect of this correlation on SCD (Essery and Pomeroy, 2004). Potential pitfalls are

incomplete modelling representations that might neglect a governing process. To capture the spatial correlations, models need to include snow redistribution, internal snowpack energetics and melt rate variability on slopes at fairly fine scales (<100 m) in complex terrain. Semi-distributed models with homogenous snow distribution over large areas or distributed models that neglect blowing snow redistribution may misrepresent spatial correlations of SWE and melt.

Another reason for models misrepresenting spatial correlations between HS0 and dHS is discussed in section 3.6, in which the mismatch of scales of dHS and HS0 patterns is discussed.

## 3.4   Variability of dHS in relation to HS0 and temporal persistence

Table 2 shows mean, standard deviation and CV values of HS and dHS in different periods and subareas. Throughout the melt season CV values of dHS were about five times smaller than those of HS. At the start of the study period, the variability of dHS was smaller than that of HS by a factor 3.7 to 6.7.

For the whole area only a weak temporal correlation (r = 0.36) was found in a pixel-by-pixel analysis between ablation patterns over the two long periods P1 and P2. Larger correlations were found for the northern subarea (r = 0.60). Ablation patterns in certain sub-periods with similar weather conditions were correlated to each other.  For instance, ablation patterns in the cool and cloudy period between 05 May and 01 June were correlated with two other rather cloudy sub-periods at the end of the study period with r = 0.49 and r = 0.64, and to the later combined period P2 (r = 0.70). Further investigation on how these correlations responded to weather was not possible given the reduced signal-to-noise ratio for shorter time periods and the inclusion of several weather types over longer periods.

## 3.5   Depletion curves

Maximum differences in dHS of up to 100% were measured (section 3.3) and were spatially persistent especially in the northern subarea. Similarly to Pomeroy et al. (2001) and Egli et al. (2012) the impact of spatial dHS on snow-cover depletion were analysed in several scenarios:

1. Variable HS0/uniform dHS: This scenario started with the measured distribution of HS at the start of the study period (HS0) and a spatially uniform dHS value was

applied for each pixel. This value was determined with observed mean ablation values shown in Table 3. Each pixel was reduced by this mean value and any negative values in HS were set to 0. SCA was defined as the ratio of the number of grid points with HS > 0 to all pixels.

2. Uniform HS0/variable dHS: In this scenario, the mean initial snow depth as shown in Table 3 was uniformly distributed in the whole snow-covered area. Spatially variable dHS values as measured with the UAV were applied to each pixel. To obtain the exact melt-out time this scenario was calculated daily using a temporally constant dHS value between flights. No exact dHS amounts were available for pixels which melted out between flights. For those pixels the mean areal dHS value was applied. The general shape of SCD curves obtained when this scenario was also calculated on the time resolution of the UAV flights.

3. Uniform HS0/uniform dHS: Similar to scenario 2, but a spatially uniform dHS value was applied to each pixel, each of which had a uniform HS0. This scenario was also calculated on a daily resolution.

In all scenarios, SCA was set to 1 for the area which was snow-covered at the start of the study period. Figure 6 shows mean HS ablation and SCD curves for the whole area and the northern subarea (top), for which more flights are available. Differences between measured development and the first scenario of uniform dHS and variable HS0 were not large. However, a large difference between measurements and the second and third scenarios with uniform HS0 and either variable or uniform dHS is obvious. Areal dHS in those scenarios was overestimated before modelled melt-out because of the overestimation of SCA. Later, areal dHS was underestimated (or zero) since most or all snow disappeared too early. This is particularly important when the aim is to model late rain-on-snow events in hydrological models (Pomeroy et al., 2016). These results indicate that it is possible to not represent the spatial melt variability in late melt and still simulate a realistic SCD curve, while this is not possible if the spatial variability of HS0 is not represented. This main feature is consistent with Egli et al. (2012).

The main reason why the observed dHS differences, which were substantial and partly persistent, did not influence SCD curves can be found in the small to negligible spatial correlation between dHS and HS0 (cf. section 3.3 and Table 1). Large correlations substantially influence SCD: Negative correlation accelerates SCD at the beginning of melt

and delays it in late melt lengthening the snowmelt season and vice versa with positive

correlations (Essery and Pomeroy, 2004).

Where correlation is insignificant, spatial melt differences can be quite large without

affecting SCD curves.  In this case, spatially variable melt can be viewed as a nearly random

process – it introduces noise into the log-normal frequency distribution of HS, but does not

affect the emergent behaviour of the SCD curve. Here, with a much larger variability of HS0

compared to dHS (see section 3.4) and only small spatial correlations between them (see

Table 1), HS0 controls the SCD.

**3.6   Scale dependencies of dHS**

Figure 7 and 8 show how the variance of dHS, the variance of explaining variables and

correlations thereof, develop with larger lag distance between point pairs (variograms and

correlograms, Eqs. 1 to 3). This gives further insights into the driving factors of ablation and

why a correlation between dHS and initial HS0 was weak in this study area during late melt.

Figure 7a, the variogram of dHS, shows that the variance increased over two distinct

length scales, one less than 50 m and one greater than 200 m. This implies that the driving

processes which generate variance for dHS need to be investigated at these two scales. In

section 3.3 a strong correlation was found between dHS and *Brightness* and *Solar Irradiance*,

but only small correlations between these and HS0. These variables were therefore analysed

with variograms and correlograms.

The variogram of *Brightness* shown in Fig. 7b indicates a variance increase only at the

small lag distances less than 50 m. This is consistent with the visual impression of a small-

scale variability of albedo shown in Fig. 3b. The correlogram shown in Fig. 7c reveals a

strong correlation of *Brightness* with dHS at these small scales ($\rho_{xy} \approx$ -0.6 at 50 m lag

distance). This demonstrates that albedo was largely responsible for the small-scale dHS

variability observed in Fig. 7a.

Figure 8a shows the variogram of *Solar Irradiance*. A small increase for length scales

less than 100 m suggests radiation and aspect differences at those scales (within-slope

variations), but the largest increase can be observed at lag distances longer than 200 m. This

scale represents slopes on both sides of the ridge and coincides with the larger scale of dHS

variance. Indeed, the correlogram (Fig. 8b) confirms that the largest correlation with dHS to

$\rho_{xy} = 0.4$ was achieved at those larger distances.

The same analysis for initial snow depth (HS0) can be seen in Fig. 8c and d. Most of the variance for snow depth is at length scales less than 100 m. The periodic behaviour shown beyond that scale may be due to the patchy snow cover which has long snow-free patches. No substantive correlation with dHS is observable on all scales (Fig. 8d).

This analysis offers further explanation why dHS and HS0 were not spatially correlated in these observations. dHS variance was related to large scale aspect changes on both slopes and medium scale albedo change, whilst snow depth was variable mainly at much smaller scales. This scale mismatch leads to a larger scatter between dHS and HS0 values and thus prevented a substantive spatial correlation.

Two processes were previously discussed and described in Fig. 5c which could drive compensating correlations between HS0 and dHS; cold content and melt energy. Cold content likely acts on a similar scale as HS0, since it depends mainly on snow depth. As shown in Fig. 5d and 5e a negative correlation driven by cold content is not uniformly present. Melt energy differences, i.e. differences in net shortwave radiation, turbulent fluxes, and net longwave radiation, are not directly dependent on snow depth, but need to spatially coincide by chance (e.g. by direction of redistribution). Acknowledging that *Solar Irradiance* is a simple proxy of melt energy, spatial coincidences between accumulation and melt energy are only present over larger distances (Fig 8b). The large scatter between HS0 and dHS results from the observation that most of the variance of HS0 occurs at much smaller scales (Fig. 8c). Figure 8d illustrates variability in the compensating correlations. At small scales below 50 m, the differences in *Solar Irradiance* are small and the cold content is responsible for slight negative correlation between HS0 and dHS. This is counteracted by *Solar Irradiance* until the distance of 250 m (cp. Fig 8a).

There needs to be a match in scaling behaviour between SWE and melt rate for these variables to develop spatial correlations. Assuming melt is primarily driven by aspect and slope differences as in the proxy *Solar Irradiance*, SWE must vary on similar scales for a correlation to develop. This may be achieved if SWE varies primarily over larger scales, e.g. in a simple topography of a ridge without gullies and with one predominant wind direction during blowing snow, in which one slope face has much larger SWE values than the other. This may also be achieved if *Solar Irradiance* acts on a smaller scale similar to HS0. This might be possible in highly complex terrain in which most slope/aspects differences can be found on scales below 100 m but this does not correspond to the "ridge" in our study site.

## 4   Conclusions and outlook

The aim of this study was to determine factors which influence areal snow ablation patterns in alpine terrain using spatially intensive observation. The dependency of snow accumulation and topographic variables with spatial melt rates were analysed for an alpine ridge in the Fortress Mountain Snow Laboratory located in the Canadian Rocky Mountains. Detailed maps of snow depth, snow depth change and snow-covered area were generated during late season ablation with UAV-based orthophotos, photogrammetry and Structure-from-Motion techniques. Snow depth and its change served as proxies for snow accumulation and melt rates. Snow depth change values were found to be spatially variable and mainly dependent on variation in solar irradiance and albedo, and likely on the cold content of the snowpack which is a function of snow depth. Local and small-scale dust variations, which had not previously been observed in the area, increased the variability of ablation.

Snowcover depletion curves were mostly dominated by the variability of initial snow depth at the start of this study rather than the variability in snow depth change. Initial snow depth variability was approximately five times larger than the variability in snow depth change in this windswept environment. The scales of variability of snow depth and snow depth change were mismatched, with snow depth variability occurring at small scales (<10 m) and snow depth change associated with the medium scale (50 m) of albedo variation or the slope scale (100s of m) of solar irradiance variation.  As a result, the initial snow depth and changes in snow depth were not strongly correlated over space, and so only initial snow depth influenced snowcover depletion.

The observations collected here show the prerequisites for strong correlations that can impact snowcover depletion curves.  Correlation between melt and snow accumulation may be driven by cold content and melt energy distributions. Whilst cold content can create a negative correlation between melt and snow accumulation, melt energy variations can create either positive or negative correlations. In order to not compensate for each other, one process needs to be dominant, or the both processes need to create a similar negative correlations.  It is also important that these variations occur at the same spatial scales.

To further investigate these arguments, longer time series of spatially detailed snowpack and snowcover observations need to be made in order to further test and examine the temporal evolution of the spatial covariance and variance of ablation and accumulation in various global alpine environments.  The results of such a study could suggest how to

parameterise snow-cover depletion and runoff models for snowmelt dominated alpine catchments, without relying on model calibration. This will help to transfer snow-hydrological models to ungauged catchments and to model future climate scenarios where snow redistribution patterns might be vastly different.

## 5  Data availability

The data is available, upon request from the database manager (Amber Peterson, in the Global Water Futures dataserver. (www.ccrnetwork.ca/outputs/data/index.php). Please refer to this website for contact details. The data involves all UAV derived grids for HS, dHS and SCA, as well as grids of explaining variables (Brightness, Deviation from North and Slope) in 1 m resolution (cp. section 2.4). Metadata is provided which explains the file naming convention of the grids (dates and variables).

## 6  Author contributions

MS collected all field data, performed postprocessing and analysis, and wrote the paper. JWP provided guidance and reviewed and revised the paper.

## 7  Competing interests

The authors declare that they have no conflict of interest.

## 8  Acknowledgement

The authors wish to acknowledge Phillip Harder for UAV training, Chris Marsh for post-processing and modelling support, May Guan and Angus Duncan for extensive field work help, Xing Fang for the assistance in modelling CHRM results, as well as Phillip Harder, Jonathan Conway, Keith Musselman, Nico Leroux and Nik Aksamit for and helpful comments and discussions. We are grateful for logistical support from Fortress Mountain Ski Resort, Cherie Westbrook for access to a differential GPS unit. Funding was provided by NSERC through Discovery and Research Tools and Instruments grants and NSERC's Changing Cold Regions Network, the Canada Research Chairs and Canada Excellence

Research Chairs programmes, Alberta Innovation, Global Water Futures, and Alberta Agriculture and Forestry. We also want to acknowledge Charles Luce and two anonymous reviewers for their constructive work on the manuscript.

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

# Figures

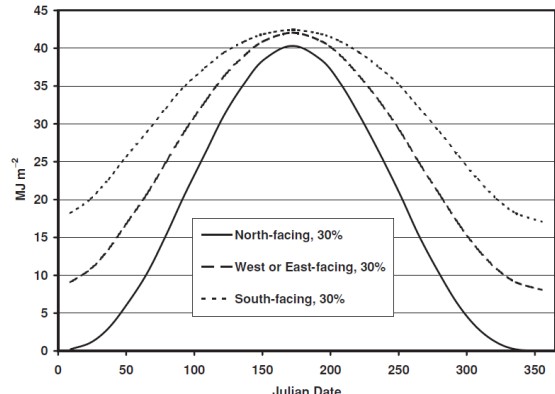

*Figure 1.  Extraterrestrial solar irradiance at 50$^o$ N for north, south and east/west facing 30 % slopes. Note the small differences as summer solstice is approached  (after Gray and Male, 1981).*

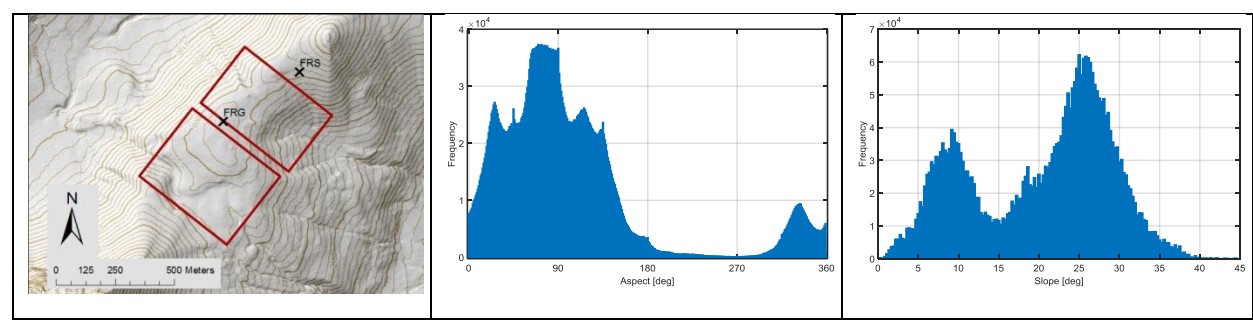

*Figure 2. Topography of the study site, (a) Overview of the two areas of investigation (red rectangles) with the location of the two weather stations (black crosses) on the alpine Fortress Ridge, Alberta, Canada, and (b) aspect distribution of the snow covered area at 27 May 2015 (spatial resolution of 10 cm, N > 3 x 10$^6$:*

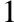

*Figure 3. UAV photogrammetric data for the study site: (a) orthomosaic from images captured on 22*
*May 2015, showing the two N and S areas of investigation (red polygons: Points indicate locations of*
*manual snow depth and differential GPS measurements over snow (red) and bare ground (green: (B)*
*enlargement of part of the study area showing evidence of dust on snow, (c) snow depth (HS) on 19*
*May 2015 and d) differences in snow depth (dHS) between that date and 1 June. The green colour in*
*(d) indicates areas excluded from analysis because of human impacts on snow or substantive*
*vegetation.*

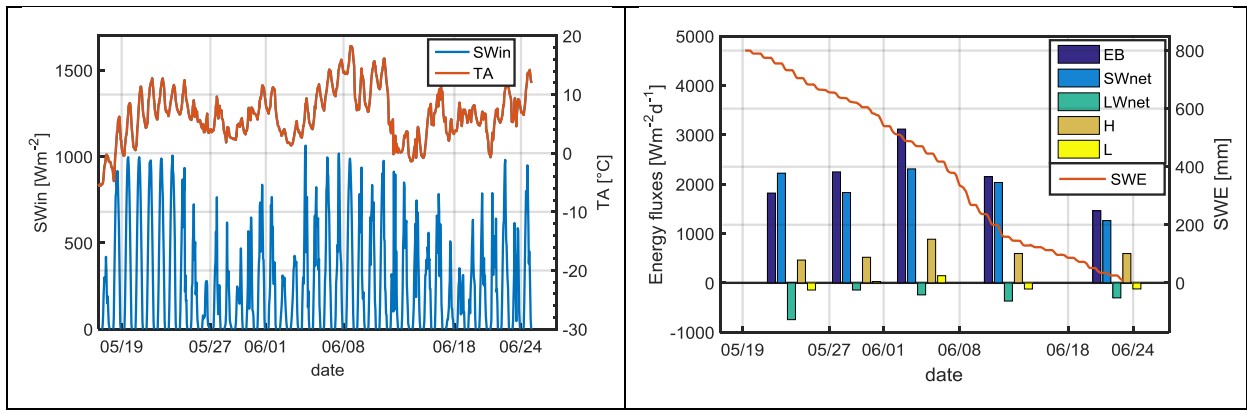

2 *Figure 4: Measured (a) and modelled (b) values at the FRG station, energy fluxes per day for periods*
3 *between UAV flights as modelled by CRHM. EB is the total energy flux, SWnet and LWnet are net*
4 *shortwave and longwave radiation, H and L are sensible and latent heat fluxes. Heat advected by rain*
5 *and ground heat flux, with only small contributions are not shown.*

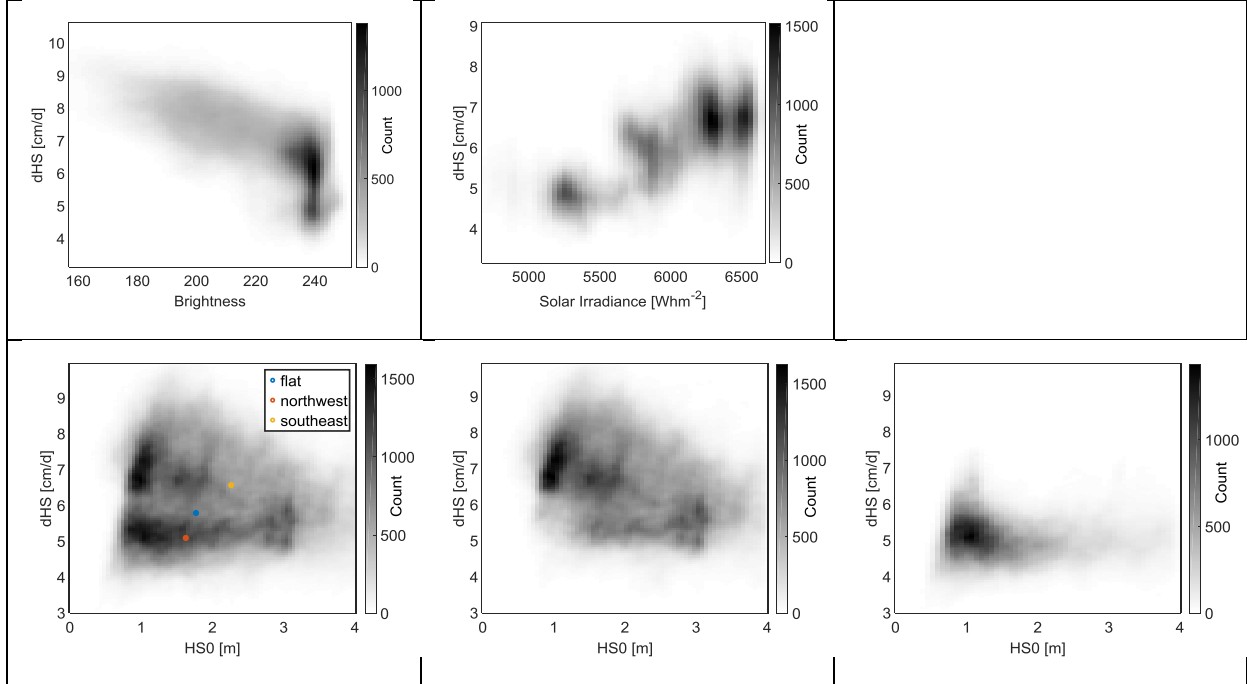

*Figure 5. Scatter plots of (a) snow brightness and b) solar irradiance versus differences in snow depth*
*(dHS) for the northern subarea during P1. Darker tones indicate a higher density of points. For (b) only*
*bright snow pixels are used (brightness > 230:  Scatter plots (c-d) show the dependence of dHS and*
*HS0 for the whole area with mean values (coloured points) of either side of the ridge and additionally*
*flat pixels (c), and only on the northwestern (d) and southeastern part of the ridge (e).*

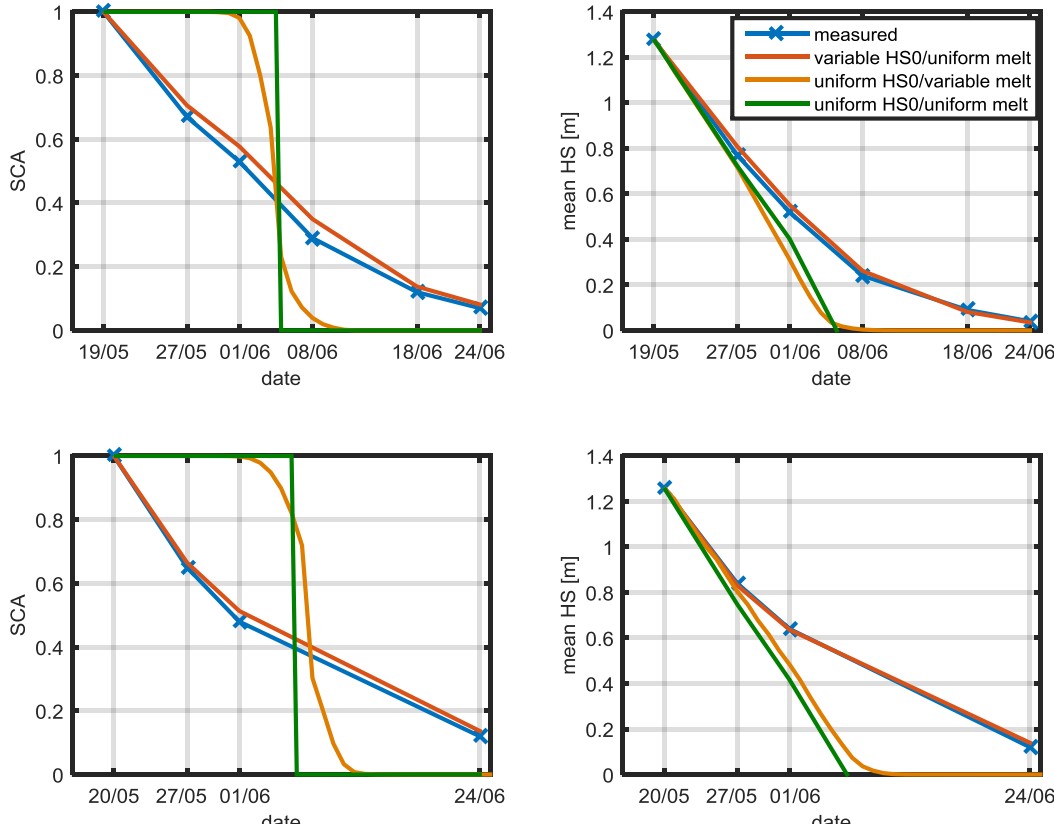

2  *Figure 6. SCD and mean HS ablation for subarea N (a, b) and the total area (c,d: Blue are measured*
3  *values, red are modelled values with initialized with measured HS distribution on May 19 and uniform*
4  *melt, green are modelled values initialized with uniform snow depth distribution and uniform melt.*

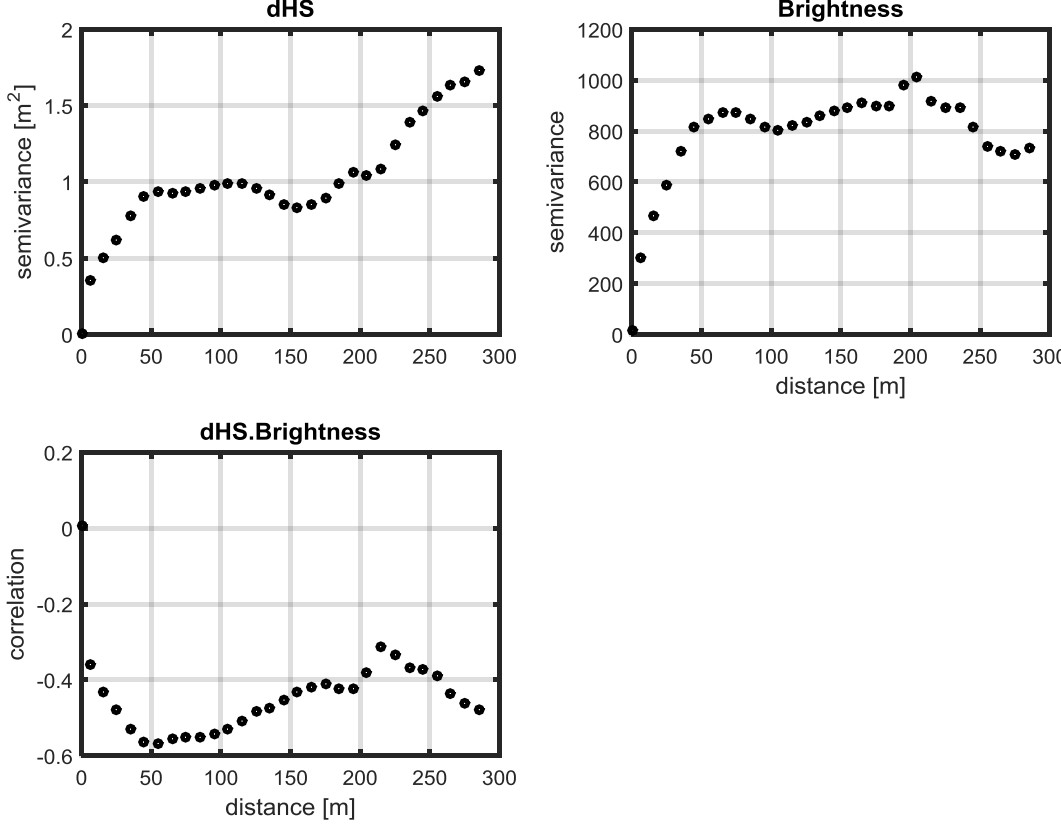

2 *Figure 7. Variograms and correlogram for dHS and Brightness.*

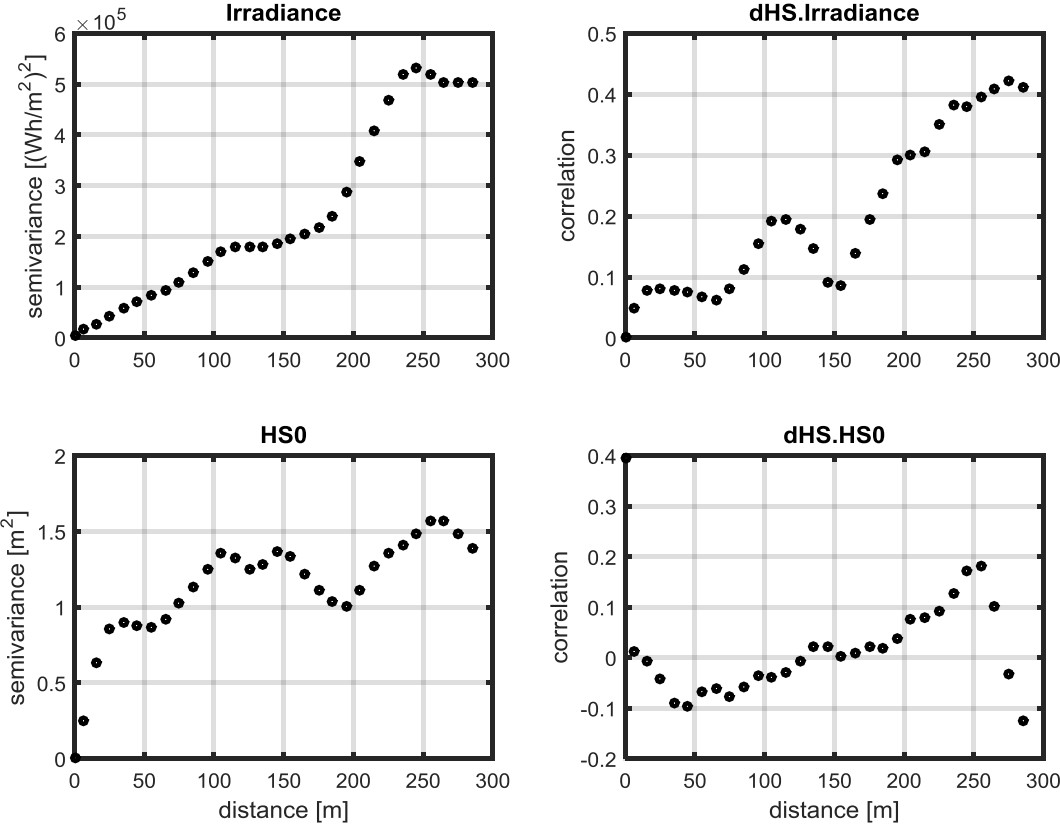

2 *Figure 8. Correlograms of dHS with modelled Irradiance and initial snow depth (HS0), respectively.*
3 *Variograms for HS0 and modelled Irradiance.*

# Tables

*Table 1. Pearson's correlations coefficient r between dHS and explaining variables. P1 is from 19 May to 01 June and P2 from 01 June to 24 June. N is number of observations.*

| Period | Area | Brightness | Degrees from North | Solar Irradiance | HS0 | N |
|--------|------|-----------|--------|--------|------|------|
| P1 | all | -0.47 | 0.56 | 0.39 | -0.12 | 3245837 |
| P2 | all | -0.59 | 0.30 | 0.01 | 0.33 | 706344 |
| P1 | N | -0.66 | 0.57 | 0.57 | -0.24 | 1410768 |
| P2 | N | 0.03 | 0.84 | 0.84 | 0.03 | 183822 |
| P1 | S | -0.43 | 0.39 | 0.33 | 0.01 | 1835069 |
| P2 | S | -0.68 | -0.21 | -0.61 | 0.30 | 522522 |

*Table 2. Mean, standard deviation (SD), Coefficient of Variation (CV) for snow depth (HS) and snow depth change (dHS) for different periods and areas. P1 was from 19 May 2015 to 01 June 2015 and P2 from 01 June 2015 to 24 June 2015. Values are given for only snow-covered areas. Values for HS are given for the start date of the period. Values for dHS are given for the area which was snow covered at the end of the melt period.*

| Period | area | HS [m] | | | dHS [cm/d] | | |
|--------|------|------|------|------|------|------|------|
| | | Mean | SD | CV | Mean | SD | CV |
| P1 | all | 1.26 | 1.16 | 0.92 | 6.22 | 1.22 | 0.20 |
| P2 | all | 1.33 | 1.13 | 0.85 | 7.57 | 1.19 | 0.16 |
| P1 | N | 1.28 | 0.93 | 0.73 | 6.86 | 1.22 | 0.18 |
| P2 | N | 0.98 | 0.73 | 0.74 | 6.76 | 1.35 | 0.20 |
| P1 | S | 1.25 | 1.27 | 1.01 | 5.72 | 0.97 | 0.17 |
| P2 | S | 1.54 | 1.27 | 0.83 | 7.86 | 0.98 | 0.12 |