# Peer review of "Processes governing snow ablation in alpine terrain."

_Hydrology and Earth System Sciences, 2018_

## Referee Comment (RC1) · Anonymous Referee #1 · 5 Jun 2018

General comments

The objective of the study is to analyse factors which control areal snow ablation and snow cover depletion in a small study area in the Canadian Rocky Mountains. The analysis is based on very detailed maps of snow depth and snow depth differences obtained by several flights of UAV in one winter season 2014/2015. The results indicate that ablation rates differed in space and were mainly related to the spatial patterns of solar irradiance and albedo. The most important factor controlling snow cover depletion was the initial distribution of SWE, which was five time s more variable than

melt variability. The authors conclude that in near summer solstice conditions the snow cover depletion curves can be calculated only from SWE spatial distribution. Generally, the topic of the manuscript is interesting and within the scope of the journal. The manuscript has a good structure and is clearly written. However, the analysis is based on only a few observations in one winter season, so the significance and generality of results are rather small. This is very well documented by the authors, who conclude that: "... clear advice to modellers is still not possible" and "Thus longer time series of spatially detailed SWE observations need to be made ...".Moreover the methodology used (UAV snow depth mapping) is not new. These facts raise a question whether the presented results provide a significantly novel contribution satisfying the HESS requirements for a scientific paper. In my opinion, the presented results are in its current form rather premature and more systematic and longer datasets are needed to justify interpretations made and to allow a transferability of results to other regions.

Specific comments

1) I found a little bit confusing connecting snow depth change directly to snow water equivalent. How valid/uncertain is the assumption of uniform snow density at 10cm spatial scale?

2) P.7, l.27: "...increase? of R2". Please check.

3) Fig. 5. Perhaps consider to switch x and Y axes (to plot dHS on Yaxis as a prediction variable). Is a simple linear relationship robust enough?

4) References: I understand that the authors wrote many papers about the subject and are expert in the field, but I feel that the references are too biased to their own work. I wonder whether all the cited works of the authors are really relevant for the topic and/or if there are some other relevant studies evaluating snow cover depletion curves and factors controlling them on different scales.

254, 2018.

---

## Referee Comment (RC2) · Anonymous Referee #2 · 13 Jun 2018

The study of Schirmer and Pomeroy used high-resolution aerial photographs of a mountain ridge to determine the spatial distribution and height of the snowpack (HS) during the melt season. Several surveys were undertaken at different times and the differences in snow height measurements (dHS) were used as proxies for ablation. The spatial patterns of ablation (dHS) was compared with pre-melt snow water equivalent (SWE, measured manually) and several topographical variables. Albedo (i.e., brightness of the snow) and solar radiation differences (i.e., deviation from North or soral irradiance) were identified at the dominant controls on dHS, whereas there was

no correlation between dHS and initial SWE. The authors explain this lack of correlation with the difference in spatial scales at which dHS and initial SWE are affected by topographic and climatic variables. The high-resolution measurements of dHS further allow to estimate the spatio-temporal variability of ablation. The authors find that the variability in ablation (dHS) is much smaller than that of initial snow depth (HS0). Consequentially, snow cover depletion curves (SCD) are less sensitive to the spatio-temporal variability of ablation and most sensitive to the HS0 of the area. The authors show this by determining and comparing SCD's from combining either uniform or variable initial HS0 with uniform or variable dHS.

The high-resolution data set of spatial snow depth distribution is unique and potentially allows interesting analyses, however, I find it difficult to identify the novel scientific contribution of this study. One of the main findings, that initial snow depth (HS0) is not correlated with changes in snow depth over time (dHS) is only briefly mentioned in Sect. 3.3. The second finding, SCD curves for the study area are largely affected by HS0 and less by dHS, has been studied extensively previously (P2L21-24, P3L31-32). I would thus recommend to revise the manuscript in a way that brings out the novelty of the authors' findings more clearly. and to help the reader to learn something. In addition, the language of the manuscript needs to be improved as some sentences are confusing and hard to understand (e.g., p11L27: "However, no study showed consistent and persistent fine-scale association between ablation and SWE suggested that they can be considered uncorrelated in modelling at fine scales."). Please find some more detailed comments below.

Introduction:

P2L30 – P3L33: It is difficult to follow the authors' train of thought here as this paragraph seems like a random collection of studies without an overarching theme that help the reader to get to the same conclusions as the authors. Is the overall point of this paragraph to show that SWE and melt are variable over time and space or that it is challenging to determine an accurate SCD curve? If so, it would help to state this as a

theme at the beginning of the paragraph.

Please explain briefly what a SCD curve is.

Methods: 2.1 Site description: Why was this study site chosen, given that the snow distribution was strongly affected by ski slopes and strong winds?

2.2 UAV Data acquisition: please don't use abbreviations in the headings or write as "Unmanned aerial vehicle (UAV) data acquisition"; How many flight were made in total? Can you please provide the dates of the individual flights in this section? Your statement on p5L20 is not clear enough: "Ideally, four flights in total were made each sampling day, two for each subarea with perpendicular flight plans. Weather conditions and technical problems often allowed only a part of this program."

2.3 Accuracy evaluation and manual measurements: From this description of the methods I understand that for each (4?) sampling day, snow depth and density (i.e., SWE) were measured at up to 7 locations over the entire field site. Were these SWE estimates assumed to be representative for the times between measurements? Did you multiply snow density with dHS to estimate the ablation rate? This is not at all clear from your statement: p6L5-9 "At these GPS measurement points, snow depth was also manually measured and snow density was measured at approximately each third of these points. Density measurements were not sufficient to confidently estimate SWE from snow depth into SWE and ablation rates from differences in snow depth. As such the originally measured quantities are analysed and interpreted as proxies for ablation and SWE in the text." Also, what do you mean by "originally measured quantities"? Please be more precise.

Results and discussion: 3.3 Spatial differences in dHS: It would be nice if you could also provide the correlation results for the remaining variable Slope, as well as the p-values for all correlations, for completeness (Table 1).

In Section 3.3. you use dHS (change in snow depth) equivalent to melt (or ablation),

although nowhere before was explained what this assumption is based on and how melt was estimated. This important bit of information only comes later (Sect. 3.4, P12L28-30); please include this description into the Methods section. Also, if you simply multiply HS and dHS with a uniform and temporally constant snow density, the variability of the resulting SWE values and melt volumes are the same as for HS and dHS multiplied by snow density.

P11L29-32: Your main finding, that is that initial HS is not correlated with dHS, is somewhat hidden in section 3.3. Given that this is a major result of your study, I would suggest to include a figure similar to Fig.5 that actually shows this lack of correlation. Also, your conclusion "These values indicate much larger SWE variability than ablation variability in this period." is equivalent to a larger variability of HS relative to dHS (in other words: the relative standard deviations of HS and dHS are same as for SWE and melt volumes). Thus, it seems confusing to use SWE instead of HS and melt instead of dHS, because SWE and melt are not measured the same way as HS and dHS.

Minor comments: - P4L5: What results of what models? A reader not familiar with snow hydrology literature has no idea what it meant by that. - P6L18: Please explain what SfM means. - P11L24: "The correlation of…" What? "… with…" - P16L22: "…varying exposures of vegetation, which is not a factor in this study." Earlier in the manuscript you state that vegetation has a strong effect on snow distribution. Please explain. - P7L27: Shouldn't it be "decrease of R2"? - P12L23: "Relative importance of ablation and initial SWE" Relative importance for what?

---

## Referee Comment (RC3) · C. Luce (Referee) · 20 Jun 2018

This paper examines whether uniform melt assumption applied to depletion curves is reasonable for a site in northern Canada. It takes a bit of reading to figure that out, but that is the essential scientific contribution being addressed.

Unfortunately, 1) it is not framed in the context of other related work showing how replication can be used well to advance the science in this particular area, and 2) there are a few questions about the statistical and sampling procedures that require

addressing. These problems could be addressed with some effort.

The most important issue is that the paper does not make a strong or compelling argument for its primary purpose or the need to replicate earlier experiments. It could be written more efficiently so that the primary scientific contribution was more prominent and readily apparent. The purpose is described in the paper as "determine factors which influence areal snow ablation patterns in alpine terrain," which is a bit vague and overarching, and the paper does not fully accomplish that task. The abstract and introduction spend most of their opening lines on the general subject of heterogeneity in snow without narrowing down to the specific issue addressed in this paper. The paper eventually goes into some depth in the introduction about depletion curves and relative contributions of melt versus accumulation variability. This is a good subject and an important subject in this field. As the authors note in P4L2-5 this is still a debate for the modeling community. An important question for the authors is why one would raise this question on Page 4 and not Page 1. Upon raising the question then, it is important to bring to bear the various answers and measurements contributing to that uncertainty already in the literature.

If better framed, the introduction should also address the need for replication of experiments on this subject in multiple places. The primary problem here is that the background material presented is by-and-large based on citations of their work or that of close colleagues. This is maybe fine for a general discourse or more obviously unique contribution. However, if one needs to make a case that more replication is needed on a subject, one needs to make a specific effort to find as much of the related literature as can be reasonably applied and explain why this particular replication is useful.

I'll pick on one citation that is already used for a different subject (general heterogeneity), but which has a nearly identical conclusion as this paper, Luce et al. 1998. We stated several times and in various ways:

"This result implies that spatial variability in snow drifting has a greater effect on the

behaviour of Upper Sheep Creek than spatial variability in solar radiation and temperature."

It would be great to discuss this and the four related papers also giving similar findings on P3L29-31 in more detail and explain why measurements in more places are useful to answer the questions brought up 3 lines later. Without some explanation (e.g. that these conclusions were derived based on only 4 sites) the lines saying that the answers are unclear following four (now five) articles that agree with each other seems almost contradictory. There is some text in the preceding page-long paragraph that describe some differences in findings, but again one has to tease out that apparently one set of findings is from forests and one from windswept areas.

I think it quite reasonable to summarize from the antecedent papers that the relative dominance of accumulation versus melt processes varies from place to place, and that adding information about another location to that list, particularly with some more detailed physical insights, could be useful. Certainly, one could bring up that there might be more value in a synthesis (e.g. along the lines of Clark et al., 2011) when trying to sort through that problem, but that requires many sites to have been sampled.

Page 19 Lines 1-4 present the key problem needing to be addressed. One would hope that the paper advances the theory and process understanding necessary to solve this problem rather than simply presenting one more example, however. It looks like there is capacity to do so with these data, but I'm not entirely certain. A well written introduction could probably narrow the subject enough that one could ask whether the finding that accumulation distributions are more important than melt distributions is a general finding for windswept sites with primarily low vegetation, or whether there are other contextual variables or information that would alter that simple generality? Alternatively, is there capacity to explore processes or causes for the lack of correlation that might otherwise be expected? For example, is the cause of low correlation a result of 1) the high sun angles during the melt, 2) dust deposition mirroring snow deposition (e.g. a process likely to cause a positive correlation between melt and accumulation

anomalies), or 3) substantially greater variability in accumulation than in melt as might be predicted from an area dominated by low slope angles and southerly and windward aspect?

At least some degree of coherent synthesis is necessary to support the addition of another paper on this subject that shows results similar to others. The heavy reliance on one or two heritages for many of the citations throughout the paper hobbles it considerably. Many of the papers cited in Clark et al., 2011 have information relevant to the discussion in this paper, and there are a number of others. There is also a need to become better acquainted with the literature. Some papers are cited for one thing when they are more relevant for another argument, or even several throughout the paper. There are also several citations in the paper (of the authors own work) that provide relatively poor support of their sentence compared to other well-known work.

Additional Points:

With respect to the analysis of correlation between HS0 and dHS, only Pearson (linear) correlation is tabulated. It would be useful to see the plots and better understand the causes for the apparent lack of correlation.

There is a great deal that should be explained about the potential effects of the sampling on the results. On P8 L17-19: In addition, one can note that the ESE wind direction is subparallel to the main ridge line. Would this have anything to do with the results? Also from Figure 2, most of the area does not look to be particularly steep, and it is by and large south facing. These do not seem like circumstances that would be likely to produce substantial variance in melt. Furthermore, most of the winds are from the south-ish, implying an expectation of relatively more scour on much of the area with only a few subdrainage/subridges causing enhanced deposition (Figure 3c) with only a little participation by the main ridge, and there mostly with slightly south facing (?) areas. And in Figure 3d, only a few areas are really analyzed. Given that areas with shallow snow tend to have more vegetation poking through (northern part of

3d), it seems like a lot of the locations with shallower initial snow are excluded from the analysis, and it is hard to sort through the impacts of that choice in finding a correlation between initial snow depth and melt rate.

On P7 L22-24: Stepwise regression is a notoriously poor method for model selection. See Burnham and Anderson (2002), for example. I would not be surprised to see similar results from a more formal model selection procedure, but it seems important to use our best understanding when applying statistics.

P2 Lines 8-10 appear to contradict lines 10-12. Lines 10-12 apply only to the special case where wind deposition occurs on multiple aspects.

P19 L5-9: I would like these authors (and, to be fair, a large number of other authors) to comment on how more time series in one place help us to transfer models to other places. This seems to be a fundamental underpinning of modern hydrological science as it is practiced, and I have not been presented with much in the way of evidence to support it.

If any further details can help clarify any of these points, please feel free to contact me. Sincerely, Charles Luce

References not already cited:

Burnham, K. P., and Anderson, D. R.: Model Selection and Multimodel Inference, Second ed., Springer, New York, 488 pp., 2002.

Clark, M. P., Hendrikx, J., Slater, A. G., Kavetski, D., Anderson, B., Cullen, N. J., Kerr, T., Hreinsson, E. Ö., and Woods, R. A.: Representing spatial variability of snow water equivalent in hydrologic and land-surface models: A review, Water Resour. Res., 47, W07539, 2011.
* * *

---

## Author Response (AR1)

Dear editor and reviewers

Many thanks for your constructive comments. We have worked on the manuscript with the following main changes:

- The introduction was newly written emphasizing on studies in complex terrain with only minor vegetation effects on snow distribution and including a larger diversity of studies to this topic
- We are now focusing more on the missing correlation between HSO and dHS as well as on explanations for this missing correlation. For this a new set of figures was included showing the correlation between HSO and dHS (Fig. 5 c-e). For the same purpose, we were also deleting some paragraphs (e.g. the comparison with other studies, which is now more concisely mentioned in the introduction, as well as the regression analysis)
- With the gained knowledge at this site we formulated prerequisites for a strong correlation between melt and SWE in other areas. With these paragraphs we think we can show that we do not show only nice data from a new study site but that we can learn something from this high-resolution data set which is useful for the development of models and the design of further investigations on this topic

**Point-by-point response to the reviews**

**Reply to reviewer 1**

We thank the reviewer for the constructive comments. We copied the reviews comments in this reply for a better readability and marked them using italic fonts.

**General comments**

The objective of the study is to analyse factors which control areal snow ablation and snow cover depletion in a small study area in the Canadian Rocky Mountains. The analysis is based on very detailed maps of snow depth and snow depth differences obtained by several flights of UAV in one winter season 2014/2015. The results indicate that ablation rates differed in space and were mainly related to the spatial patterns of solar irradiance and albedo. The most important factor controlling snow cover depletion was the initial distribution of SWE, which was five time s more variable than melt variability. The authors conclude that in near summer solstice conditions the snow cover depletion curves can be calculated only from SWE spatial distribution. Generally, the topic of the manuscript is interesting and within the scope of the journal. The manuscript has a good structure and is clearly written. However, the analysis is based on only a few observations in one winter season, so the significance and generality of results are rather small. This is very well documented by the authors, who conclude that: "... clear advice to modellers is still not possible" and "Thus longer time series of spatially detailed SWE observations need to be made ...".

Moreover the methodology used (UAV snow depth mapping) is not new.

We agree with the reviewer that there are papers on UAV snow depth mapping, but mainly on the accuracy of this method. This is one of the first using this methodology to answer snow-hydrological questions.

These facts raise a question whether the presented results provide a significantly novel contribution satisfying the HESS requirements for a scientific paper. In my opinion, the presented results are in its current form rather premature and more systematic and longer datasets are needed to justify interpretations made and to allow a transferability of results to other regions.

The novelty is given with the high-resolution data set. Since snow depth is known to vary largely below a scale break of tens of meters in alpine environments we think that a replication of studies which relied on manual probing in a much coarser resolution is needed. Manual probing with a spacing at or larger this scale break may complicate the interpretation of the results (Clark et al., 2011), since the dominant spatial structure can hardly be captured. A similar high resolution data set was only presented by Grünewald et al. (2010) and Egli et al. (2012), which only covers one study region and two different seasons. A replication of these studies to other areas is urgently needed to show the transferability of results. Furthermore, we present an explanation on the missing correlation between SWE and melt. This is now more precisely mentioned in a more focused way in the revised manuscript.

While we feel that this paper makes a significant movement forward in our understanding of the relationships between snow accumulation and snowmelt patterns at multiple scales, final answers can only be given if more replications like this study are available. Given the large effort in field data acquisition and processing of this high-resolution data set to good quality levels (the signal of a melt period needs to overcome the noise) and the rare availability of long melt events without snowfall, this can only be a stepwise process. However, this study may initiate more replications with indicating the urgent need for this to better guide snow-hydrological model design.

During our field work in 2015 there was nearly no knowledge of the spatial noise inherent to this method. Thus, we ended up with more field days than in the final paper (9 days instead of 5). Since the noise is site and weather dependent, as well as the signal, i.e. the melt amounts, also in 2018 we would probably include much more flight days than what can be finally use. Post-processing of one field day can be done within one day, however, only if the data is acceptable with standard settings. In our case, it took us several days to achieve acceptable data quality with changing post-processing settings. This shows the tremendous effort to achieve good results, which implies that single studies using this technology can only add to existing knowledge.

We formulated prerequisites for a strong correlation between melt and SWE in other areas. With these paragraphs we think we discussed the transferability of these results to other regions.

1) I found a little bit confusing connecting snow depth change directly to snow water equivalent. How valid/uncertain is the assumption of uniform snow density at 10cm spatial scale?

First of all we only use HS and dHS in the final manuscript. However, this still implies that they can be used as proxies for SWE and melt. In the newly written introduction we point to other studies which rely on spatial model results. Spatial modelling of snowmelt in complex terrain inherits a large amount of uncertainty. For example, energy balance modelling relies on assessment of turbulent fluxes, which is dependent on local wind fields. Those wind fields in complex terrain are very difficult to estimate (e.g. Mott et al., 2010; Musselman et al., 2015). To be independent of model uncertainties, we chose semi-direct measurements of melt and SWE. These also have uncertainties, e.g. one has to either apply no density (depth only) or just a few density measurements or modelled densities. However, it is well known that snow depth varies to a much larger degree than density (e.g. Pomeroy and Gray, 1995; López-Moreno et al., 2013). As a consequence, it is common to estimate areal SWE with a small number of representative density measurements and a high number of snow depth data (e.g. Steppuhn and Dyck, 1974; Pomeroy and Gray, 1995; Rovansek et al., 1993; Elder et al., 1998; Jonas et al., 2009). We were able to take a few SWE measurements, but we did not multiply dHS with snow density to estimate the ablation rate, because we think that our SWE measurements were neither representative for the whole area nor for the time between the measurements. This is why we analysed and interpreted HS and dHS as proxies of SWE and ablation rate.

This does not answer your question on snow density variations at a 10 cm scale. There are very few measurements assessing the density variability at smaller scales. For example, Proksch et al. (2015) showed SnowMicroPen measurements along a transect with 0.5 m spacing in the Antarctica (cp. their Fig.

12), which shows density variations resulting from different deposition and metamorphic processes. Given this lack of knowledge, we think that in our study region, which has large differences in HS and rather deep snowpacks, density differences are mainly driven by larger scale deposition processes (~tens of meters) rather than smaller scale metamorphic processes driven by e.g. summer terrain or vegetation differences as it may be the case in shallow snowpacks. Thus, we suggest that the results of López-Moreno et al. (2013) also apply in our study area at the 10 cm scale as an increase of variability of density at this scale seems unlikely.

**2) P.7, I.27: ". . .increase? of R2". Please check.**

Many thanks for finding this error. However, we decided on excluding the topic stepwise regression in the revised version.

**3) Fig. 5. Perhaps consider to switch x and Y axes (to plot dHS on Yaxis as a prediction variable). Is a simple linear relationship robust enough?**

We will switch the axis and delete the regression lines. A new Figure 5 including the relationship to HS0 was requested by reviewer 2.

4) References: I understand that the authors wrote many papers about the subject and are expert in the field, but I feel that the references are too biased to their own work. I wonder whether all the cited works of the authors are really relevant for the topic and/or if there are some other relevant studies evaluating snow cover depletion curves and factors controlling them on different scales.

The introduction has been changed substantially following this comment and the suggestions of reviewer 3. This will include a diversification of cited authors, as well as focusing on relevant work for this topic.

**References**

Elder, Kelly, Walter Rosenthal, and Robert E. Davis. "Estimating the spatial distribution of snow water equivalence in a montane watershed." Hydrological Processes 12, no. 10-11 (1998): 1793-1808.

Jonas, Tobias, Christoph Marty, and Jan Magnusson. "Estimating the snow water equivalent from snow depth measurements in the Swiss Alps." Journal of Hydrology 378, no. 1-2 (2009): 161-167.

López-Moreno, Juan I., S. R. Fassnacht, J. T. Heath, K. N. Musselman, Jesús Revuelto, J. Latron, Enrique Morán-Tejeda, and Tobias Jonas. "Small scale spatial variability of snow density and depth over complex alpine terrain: Implications for estimating snow water equivalent." Advances in water resources 55 (2013): 40-52.

Musselman, Keith N., John W. Pomeroy, Richard LH Essery, and Nicolas Leroux. "Impact of windflow calculations on simulations of alpine snow accumulation, redistribution and ablation." *Hydrological Processes* 29, no. 18 (2015): 3983-3999.

Pomeroy, J. W., and D. M. Gray. "Snowcover accumulation, relocation and management." *Bulletin of the International Society of Soil Science no* 88, no. 2 (1995).

Proksch, Martin, Henning Löwe, and Martin Schneebeli. "Density, specific surface area, and correlation length of snow measured by high-resolution penetrometry." Journal of Geophysical Research: Earth Surface 120, no. 2 (2015): 346-362.

Rovansek, R. J., D. L. Kane, and L. D. Hinzman. "Improving estimates of snowpack water equivalent using double sampling." In Proceedings of the 61st Western Snow Conference, pp. 157-163. 1993.

Steppuhn, H., and G. E. Dyck. "Estimating true basin snowcover." In Advances Concepts and Techniques in the Study of Snow and Ice Resources, Asilomar Conference Grounds, Monterey, CA, USA. Washington, DC: Natl. Acad. Of Sci., 1974.

**Reply to reviewer 2**

We thank the reviewer for the constructive comments. We copied the reviews comments in this reply for a better readability and marked them using italic fonts.

The study of Schirmer and Pomeroy used high-resolution aerial photographs of a mountain ridge to determine the spatial distribution and height of the snowpack (HS) during the melt season. Several surveys were undertaken at different times and the differences in snow height measurements (dHS) were used as proxies for ablation. The spatial patterns of ablation (dHS) was compared with pre-melt snow water equivalent (SWE, measured manually) and several topographical variables. Albedo (i.e., brightness of the snow) and solar radiation differences (i.e., deviation from North or soral irradiance) were identified at the dominant controls on dHS, whereas there was no correlation between dHS and initial SWE. The authors explain this lack of correlation with the difference in spatial scales at which dHS and initial SWE are affected by topographic and climatic variables. The high-resolution measurements of dHS further allow to estimate the spatio-temporal variability of ablation. The authors find that the variability in ablation (dHS) is much smaller than that of initial snow depth (HS0). Consequentially, snow cover depletion curves (SCD) are less sensitive to the spatiotemporal variability of ablation and most sensitive to the HS0 of the area. The authors show this by determining and comparing SCD's from combining either uniform or variable initial HS0 with uniform or variable dHS.

The high-resolution data set of spatial snow depth distribution is unique and potentially allows interesting analyses, however, I find it difficult to identify the novel scientific contribution of this study. One of the main findings, that initial snow depth (HS0) is not correlated with changes in snow depth over time (dHS) is only briefly mentioned in Sect. 3.3.

This and finding an explanation for the lack of correlation is a focus in the revised manuscript.

The second finding, SCD curves for the study area are largely affected by HS0 and less by dHS, has been studied extensively previously (P2L21-24, P3L31-32). I would thus recommend to revise the manuscript in a way that brings out the novelty of the authors' findings more clearly and to help the reader to learn something.

The novelty is given with the high-resolution data set. Since snow depth is known to vary largely below a scale break of tens of meters in alpine environments we think that a replication of studies which relied on much coarser manual probing is needed. Manual probing with a spacing at or larger this scale break may complicate the interpretation of the results (Clark et al., 2011), since the dominant spatial structure can hardly be captured. A high resolution data set was presented by Grünewald et al. (2010) and Egli et al. (2012), but only covered one study region and two different seasons. Testing of these studies in other areas is urgently needed to show the transferability of results. Furthermore, we present an explanation on the missing correlation between SWE and melt. This is now more precisely mentioned in a more focused way in the revised manuscript.

In addition, the language of the manuscript needs to be improved as some sentences are confusing and hard to understand (e.g., p11L27: "However, no study showed consistent and persistent fine-scale association between ablation and SWE suggested that they can be considered uncorrelated in modelling at fine scales."). Please find some more detailed comments below.

We have revised the language throughout the manuscript.

Introduction:

P2L30 – P3L33: It is difficult to follow the authors' train of thought here as this paragraph seems like a random collection of studies without an overarching theme that help the reader to get to the same conclusions as the authors. Is the overall point of this paragraph to show that SWE and melt are variable over time and space or that it is challenging to determine an accurate SCD curve? If so, it would help to state this as a theme at the beginning of the paragraph.

We have re-written the introduction to focus on studies with minor vegetation effects on snow depth distribution and on found correlations between SWE and melt.

Please explain briefly what a SCD curve is.

Done (p. 2 l. 19f).

Methods: 2.1 Site description: Why was this study site chosen, given that the snow distribution was strongly affected by ski slopes and strong winds?

We have written a better explanation on why we chose this study area (new section 2.1). The strong winds were the attractive point of this location. The skiers influence was small and spatially limited so that impacted zones could be excluded in the study area.

2.2 UAV Data acquisition: please don't use abbreviations in the headings or write as "Unmanned aerial vehicle (UAV) data acquisition"; How many flight were made in total? Can you please provide the dates of the individual flights in this section?

Done as suggested. The UAV was flown 18 times over snow from 15 May to 24 June 2015 at eight different days with substantial depth differences between these days and four flights over bare ground on 24 July 2015. However, as stated in the manuscript, we had to restrict analysis to two melt periods.

Your statement on p5L20 is not clear enough: "Ideally, four flights in total were made each sampling day, two for each subarea with perpendicular flight plans. Weather conditions and technical problems often allowed only a part of this program."

We have clarified this topic and now mention why perpendicular matter, why subareas were defined, and what weather conditions and technical problems restricted the surveys.

2.3 Accuracy evaluation and manual measurements: From this description of the methods I understand that for each (4?) sampling day, snow depth and density (i.e., SWE) were measured at up to 7 locations over the entire field site. Were these SWE estimates assumed to be representative for the times between measurements? Did you multiply snow density with dHS to estimate the ablation rate? This is not at all clear from your statement: p6L5-9 "At these GPS measurement points, snow depth was also manually measured and snow density was measured at approximately each third of these points. Density measurements were not sufficient to confidently estimate SWE from snow depth into SWE and ablation rates from differences in snow depth. As such the originally measured quantities are analysed and interpreted as proxies for ablation and SWE in the text." Also, what do you mean by "originally measured quantities"? Please be more precise.

We have clarified this in the revised manuscript. We did not multiply dHS with snow density to estimate the ablation rate, because we think that our SWE measurements were neither representative for the whole area nor for the time in between the measurements. This is why we analysed and interpreted HS and dHS as proxies SWE and ablation rate.

Results and discussion: 3.3 Spatial differences in dHS: It would be nice if you could also provide the correlation results for the remaining variable Slope, as well as the *p*-values for all correlations, for completeness (Table 1).

We have included slope. P-values are not very meaningful for these large numbers of observations since statistical significance is almost always achieved.

In Section 3.3. you use dHS (change in snow depth) equivalent to melt (or ablation), although nowhere before was explained what this assumption is based on and how melt was estimated. This important bit of information only comes later (Sect. 3.4, P12L28-30); please include this description into the Methods section. Also, if you simply multiply HS and dHS with a uniform and temporally constant snow density, the variability of the resulting SWE values and melt volumes are the same as for HS and dHS multiplied by snow density.

We used the words melt and SWE for a better readability in the original submission, lines P12L28-P13L2. However, in the revised manuscript we will only use the "words" dHS and HS.

P11L29-32: Your main finding, that is that initial HS is not correlated with dHS, is somewhat hidden in section 3.3. Given that this is a major result of your study, I would suggest to include a figure similar to Fig.5 that actually shows this lack of correlation.

This is done as suggested.

Also, your conclusion "These values indicate much larger SWE variability than ablation variability in this period." is equivalent to a larger variability of HS relative to dHS (in other words: the relative standard deviations of HS and dHS are same as for SWE and melt volumes). Thus, it seems confusing to use SWE instead of HS and melt instead of dHS, because SWE and melt are not measured the same way as HS and dHS.

In the revised manuscript we only use HS and dHS.

Minor comments:

- P4L5: What results of what models? A reader not familiar with snow hydrology literature has no idea what it meant by that.

We have re-written the introduction to address this.

- P6L18: Please explain what SfM means.

The abbreviation SfM was explained in P4L15.

- P11L24: "The correlation of. . ." What? ". . . with. . ."

This is included in the revised manuscript. The correlation of dHS with ...

**- P16L22:**

"...varying exposures of vegetation, which is not a factor in this study." Earlier in the manuscript you state that vegetation has a strong effect on snow distribution. Please explain.

We have indeed written in P4L29ff that vegetation had an effect on snow distribution, but that these areas were exluded from the analysis. We will change the wording to clarify this topic.

- P7L27: Shouldn't it be "decrease of R2"?

Yes, many thanks for finding this error.

- P12L23: "Relative importance of ablation and initial SWE" Relative importance for what?

Many thanks, we will change this into Relative importance of dHS and HS0 on snow cover depletion.

**Reply to reviewer Charles Luce**

We thank Charles Luce for his detailed review.

This paper examines whether uniform melt assumption applied to depletion curves is reasonable for a site in northern Canada. It takes a bit of reading to figure that out, but that is the essential scientific contribution being addressed.

Unfortunately, 1) it is not framed in the context of other related work showing how replication can be used well to advance the science in this particular area, and 2) there are a few questions about the statistical and sampling procedures that require addressing. These problems could be addressed with some effort. The most important issue is that the paper does not make a strong or compelling argument for its primary purpose or the need to replicate earlier experiments. It could be written more efficiently so that the primary scientific contribution was more prominent and readily apparent. The purpose is described in the paper as "determine factors which influence areal snow ablation patterns in alpine terrain," which is a bit vague and overarching, and the paper does not fully accomplish that task. The abstract and introduction spend most of their opening lines on the general subject of heterogeneity in snow without narrowing down to the specific issue addressed in this paper. The paper eventually goes into some depth in the introduction about depletion curves and relative contributions of melt versus accumulation variability. This is a good subject and an important subject in this field. As the authors note in P4L2-5 this is still a debate for the modeling community. An important question for the authors is why one would raise this question on Page 4 and not Page 1. Upon raising the question then, it is important to bring to bear the various answers and measurements contributing to that uncertainty alreadv in the literature.

If better framed, the introduction should also address the need for replication of experiments on this subject in multiple places. The primary problem here is that the background material presented is by-and-large based on citations of their work or that of close colleagues. This is maybe fine for a general discourse or more obviously unique contribution. However, if one needs to make a case that more replication is needed on a subject, one needs to make a specific effort to find as much of the related literature as can be reasonably applied and explain why this particular replication is useful. I'll pick on one citation that is already used for a different subject (general heterogeneity), but which has a nearly identical conclusion as this paper, Luce et al. 1998. We stated several times and in various ways:

"This result implies that spatial variability in snow drifting has a greater effect on the behaviour of Upper Sheep Creek than spatial variability in solar radiation and temperature." It would be great to discuss this and the four related papers also giving similar findings on P3L29-31 in more detail and explain why measurements in more places are useful to answer the questions brought up 3 lines later. Without some explanation (e.g. that these conclusions were derived based on only 4 sites) the lines saying that the answers are unclear following four (now five) articles that agree with each other seems almost contradictory. There is some text in the preceding page-long paragraph that describe some differences in findings, but again one has to tease out that apparently one set of findings is from forests and one from windswept areas.

I think it quite reasonable to summarize from the antecedent papers that the relative dominance of accumulation versus melt processes varies from place to place, and that adding information about another location to that list, particularly with some more detailed physical insights, could be useful. Certainly, one could bring up that there might be more value in a synthesis (e.g. along the lines of Clark et al., 2011) when trying to sort through that problem, but that requires many sites to have been sampled. Page 19 Lines 1-4 present the key problem needing to be addressed. One would hope that the paper advances the theory and process understanding necessary to solve this problem rather than simply presenting one more example, however. It looks like there is capacity to do so with these data, but I'm not entirely certain. A well written introduction could probably narrow the subject enough that one could ask whether the finding that accumulation distributions are more important than melt distributions is a general finding for windswept sites with primarily low vegetation, or whether there are other contextual variables or information that would alter that simple generality? Alternatively, is there capacity to explore processes or causes for the lack of correlation that might otherwise be expected? For example, is the cause of low correlation a result of 1) the high sun angles during the melt, 2) dust deposition mirroring snow deposition (e.g. a process likely to cause a positive correlation between melt and accumulation anomalies), or 3) substantially greater variability in accumulation than in melt as might be predicted from an area dominated by low slope angles and southerly and windward aspect?

At least some degree of coherent synthesis is necessary to support the addition of another paper on this subject that shows results similar to others. The heavy reliance on one or two heritages for many of the citations throughout the paper hobbles it considerably. Many of the papers cited in Clark et al., 2011 have information relevant to the discussion in this paper, and there are a number of others. There is also a need to become better acquainted with the literature. Some papers are cited for one thing when they are more relevant for another argument, or even several throughout the paper. There are also several citations in the paper (of the authors own work) that provide relatively poor support of their sentence compared to other well-known work.

With respect to the analysis of correlation between HS0 and dHS, only Pearson (linear) correlation is tabulated. It would be useful to see the plots and better understand the causes for the apparent lack of correlation.

We have re-written the introduction to implement the suggestions by this reviewer. For example, we have narrowed the introduction to focus on alpine studies with primarily low vegetation. We added other related work as this reviewer suggested and discussed their findings in more detail in the introduction. We emphasized why a replication at this site is meaningful and instructive. The additional benefit of this contribution is now more clearly written: The novelty is shown in using high-resolution data set to permit multiscale analysis. Since snow depth is known to vary mostly below a scale break of tens of meters in alpine environments we think that testing previous coarse resolution manual probing based studies with high resolution observations was needed. Manual probing with a spacing at or larger this scale break may complicate the interpretation of the results (Clark et al., 2011), since the dominant spatial structure can hardly be captured. Similar high resolution datasets have only been presented by Grünewald et al. (2010) and Egli et al. (2012), and they covered only one study region. Testing in other areas was urgently needed to show the transferability of results. Furthermore, we present a novel explanation on the missing correlation between SWE and melt. This is now more precisely mentioned in a more focused way in the revised manuscript.

Grünewald et al. (2010) and Egli et al. (2012) were not able to provide an explanation about the lack of correlation between HS and dHS. We now contribute with an explanation why uniform melt is applicable at multiple scales at this site in southern Canada. Doing this we explored the causes of the lack of correlation as Charles Luce requested. The observed scale difference between melt and SWE prohibits a large correlation between both. Snow depth varies in this area on smaller scales than melt differences driven by aspect differences, which are the typical melt energy differences included in spatial models. Only small scale albedo differences (quite untypical for this and other areas) were responsible for small scale variations in melt. The short scale break in snow depth has been reported by other studies as well. The open question is if melt is in general a spatially much smoother field than HS in alpine areas. This can only be answered if more high resolution studies become available in other mountain regions to confirm the strong results shown here. Given the large effort in field data acquisition and processing of this high-resolution data set to a good quality (the signal of melt periods needs to overcome the noise) and the rare availability of long melt events without snowfall, this can only be a stepwise process. However, this study may initiate more replications by indicating the urgent need for this to better guide snow-hydrological model design.

To better focus on the lack of correlation between melt and SWE we changed the results section and included a figure showing the lack of correlation between HS0 and dHS. We also deleted parts less relevant for this main conclusion, e.g. the stepwise regression results.

There is a great deal that should be explained about the potential effects of the sampling on the results. On P8 L17-19: In addition, one can note that the ESE wind direction is subparallel to the main ridge line. Would this have anything to do with the results?

Also from Figure 2, most of the area does not look to be particularly steep, and it is by and large south facing. These do not seem like circumstances that would be likely to produce substantial variance in melt.

The wind directions varied widely and sometimes where perpendicular and sometimes parallel to the ridgeline – that variability is a characteristic of this region as it is subject to westerly Chinooks, cold northerly flows and wet upslope flows from the east. The area includes an initial snowcovered area which is steep with varying aspects, although the east aspect is overrepresented (Fig 2b). The two large drifts visible in Figure 3d have a southerly component and are over 30 degrees steep. The Northwest facing slopes of the ridge are similarly steep. Flat parts are on top of the ridge which is only partly snowcovered, mostly in the southern part of the study area. In contrast to Grünewald et al. (2010) we observed spatial melt differences to be dependent on aspect and slope (Figure 5b). We have included Figure 2c showing the slope distribution of the initially snow covered area to make this more clear.

**Furthermore, most of the winds are**

from the south-ish, implying an expectation of relatively more scour on much of the area with only a few subdrainage/subridges causing enhanced deposition (Figure 3c) with only a little participation by the main ridge, and there mostly with slightly south facing (?) areas.

We do not fully understand this point. The winds were from the north and the south and also along the ridge. The south face had massive snow drifts and was not scoured.

And in Figure 3d, only a few areas are really analyzed. Given that areas with shallow snow tend to have more vegetation poking through (northern part of 3d), it seems like a lot of the locations with shallower initial snow are excluded from the analysis, and it is hard to sort through the impacts of that choice in finding a correlation between initial snow depth and melt rate.

We excluded vegetation effects independently from snow depth appearance, for the included area vegetation played no role as it was bare ground before the vegetation period allowed to grow a few centimeters of alpine grass. This is now more clearly stated. There were many sites with shallow snow to begin with.

On P7 L22-24: Stepwise regression is a notoriously poor method for model selection. See Burnham and Anderson (2002), for example. I would not be surprised to see similar results from a more formal model selection procedure, but it seems important to use our best understanding when applying statistics.

We agree with the reviewer to apply more appropriate statistical methods. Following the reviewers suggestions to focus on the main results, we have deleted this paragraph.

P2 Lines 8-10 appear to contradict lines 10-12. Lines 10-12 apply only to the special case where wind deposition occurs on multiple aspects.

We have clarified this topic.

P19 L5-9: I would like these authors (and, to be fair, a large number of other authors) to comment on how more time series in one place help us to transfer models to other places. This seems to be a fundamental underpinning of modern hydrological science as it is practiced, and I have not been presented with much in the way of evidence to support it.

In Fortress Mountain there are only a few papers existent as this is a rather new study site and no previous papers have studied this topic there. We also try to discuss the potential to extrapolate our results. Moreover, as discussed above, we see this study as an initiation of new multiscale studies in other areas where airborne snow depth data is available.

**Factors influencing spring and summer arealProcesses governing snow ablation and snowcover depletion in alpine terrain: detailed. Detailed measurements from the Canadian Rockies**

Michael Schirmer1,2 and John W. Pomeroy1

[1] Centre for Hydrology, University of Saskatchewan, Canada

[2] WSL Institute for Snow and Avalanche Research SLF, Davos, Switzerland

Correspondence to: Michael Schirmer (michael.schirmer@slf.ch)

**11 Abstract**

The spatial distribution of snow water equivalent (SWE) and melt are important to 13 estimating areal melt rates and snowcover depletion dynamics but are rarely measured in 14 detail during the late ablation period. This study contributes the result of results from high 15 resolution observations made using large numbers of sequential aerial photographs taken from 16 an Unmanned Aerial Vehicle on an alpine ridge in the Fortress Mountain Snow Laboratory in 17 the Canadian Rocky Mountains from May to July. With Structure-from-Motion and 18 thresholding techniques, spatial maps of snow depth, snowcover and differences in snow 19 depth (dHS) during ablation were generated in very high resolution as proxies for spatial 20 SWE, spatial ablation rates, and snowcover depletion (SCD). The results indicate that the 21 initial distribution of SWEsnow depth was highly variable due to overwinter snow 22 redistribution and the subsequent distribution of ablation rates dHS was also variable due to 23 albedo, slope/aspect and other unaccountable differences. However, the initial distribution of 24 SWEsnow depth was five times more variable than that of subsequent ablation rates, even 25 though ablation differences were substantial, with variabilitydHS values which varied by a 26 factor of two between north and south aspects, and dHS patterns were somewhat spatially 27 persistent over time. Ablation rate patterns but had an insubstantial impact on SCD curves, 28 which were overwhelmingly governed by the initial distribution of SWEsnow depth. The 29 reasons for this are that variations in irradiance to slopes on north and south aspects in the near-summer solstice period are relatively small and only a weak spatial correlation 1 2 developed between initial SWEsnow depth and ablation ratesdHS. Previous research has shown that spatial correlations between SWE and ablation rates can strongly influence SCD 3 The results presented here are in contrast to alpine, shrub tundra and forest 4 curves. 5 observations taken during ablation at higher latitudes and/or earlier in spring but in agreement with other near-summer observations in alpine environments. Whilst variations in net solar 6 7 irradiance to snow were due to small scale variations in localized dust deposition from eroded 8 ridgetop soil and larger scale differences in slope and aspect, variations in SWE were due to 9 intense over winter blowing snow storms with deposition from multiple directions of snow transport to incised gullies and slope breaks. This condition differs considerably from 10 11 situations where wind transport from primarily one direction leads to preferential SWE 12 loading on slopes of particular aspects, which can lead to a spatial correlation between SWE 13 and ablation rate. These findings suggest that in near-summer solstice conditions and environments where snow redistribution is substantial, then mountain SCD curves can be 14 15 calculated using the spatial distribution of SWE alone, and that hydrological and atmospheric models need to implement a realistic distribution of SWE in order to do this Analysing the 16 reasons for a missing correlation in this study area provided some prerequisites for large 17 spatial correlations and for when these need to be taken into account by SCD curves. These 18 19 findings suggest that hydrological and atmospheric models need to incorporate realistic 20 distributions of SWE, melt energy and cold content and so must account for correlations between SWE and melt in order to accurately model snowcover depletion. 21

**23 **1** Introduction**

The spatial variability of snow water equivalent (SWE) during melt exerts an 25 important control on catchment or grid-scale meltwater generation-averaged snowmelt (Pomeroy et al., 1998). Terrain induced; Liston, 1999). When focussing on complex terrain 26 27 with only minor vegetation effects on SWE distribution, differences in precipitation-and terrain and vegetation induced differences in, snow redistribution, melt energy and freezing 28 29 levels lead to a spatially variable distribution of SWE (e.g. Clark et al., 2011). For modellers 30 of snow-hydrological applications the question arises as to which of those processes need to be considered. It is well known that south-facing slopes receive more melt energy than do 31 32 north-facing slopes due to differences in solar radiation. At 50°N on April 1, the differences are already 40% for a slightly inclined slope of 10°, however, these differences decrease as 1 2 summer solstice approaches (Figure 1, Gray and Male, 1981). It is also well known that SWE 3 distribution at the start of the melting season. In many environments peak accumulation is highly variable in alpine terrain. Liston et al. (2004) presented maps with regional differences 4 5 of coefficients of variation (CV) of snow depth. For alpine regions a CV of 0.85 is suggested. Both, the variability in melt energy and SWE influence snow cover depletion. This can be 6 7 visualized in snow-cover depletion curves, which are a function of snow-covered area (SCA) 8 over time or grid-averaged SWE (e.g. Essery and Pomeroy, 2004; Clark et al., 2011). Both 9 studies illustrate with theoretical simulations how increasing melt rate and peak SWE 10 variability change the rate of areal snow-cover depletion. From their theoretical illustrations 11 (Fig. 3 and 4 in Clark et al., 2011), it is clear that in alpine regions with a large variability in 12 melt rate and peak SWE, ignoring SWE rather than melt rate variability would be the greater 13 modelling mistake. However, as Pomeroy et al. (2004) pointed out, the importance of melt 14 variability on SCD increases if a spatial correlation between melt and SWE exists. This 15 suggests that in alpine terrain the question of relative contribution of spatially variable melt rates or snow redistribution on SCD can be reduced to the question of whether such a 16 17 correlation between melt and SWE exists and how large it is.

Besides theoretical considerations, there are a number of existing field and modelling studies on the relative importance of spatially variable melt or snow redistribution on SCD. 19 There are studies have found the temporal progression of snow-cover depletion (SCD) has 20 21 been found to be governed primarily by the premelt distribution of SWE variability caused by 22 snow redistribution rather than the variability caused by melt rate differences (Shook and Gray, 1996; Donald et al., 1995; Marsh and Pomeroy, 19961994; Luce et al., 1998, 1999; 23 Pomeroy et al. 1998; Pomeroy et al. 2001; Anderton, 2004; Egli et al., 2012). These studies 24 25 show with spatial observations that snow cover depletion (SCD) can be modelled with statistics derived during peak accumulation. Luce et al. (1998) modelled snow cover depletion 26 27 with a spatially distributed energy balance model which integrated drifting snow 28 redistribution based on an empirically derived drift factor. Ignoring this drift factor 29 deteriorated model results, which suggests the relative importance of snow redistribution over melt variability. Grünewald et al. All of these studies have explained the observations by 30 31 (2010) made indirect measurements of spatial melt rate and SWE via snow depth (HS) and 32 the change in HS (dHS) by terrestrial LiDAR by and applying a few measured bulk densities to estimate SWE and ablation rates. They found that SWE and melt rate were spatially 33

uncorrelated over most of their ablation season, except for a correlation coefficient of r = -0.41 2 for one sub period. They noted that the variability of SWE was much larger than the 3 variability of melt rates. In the same study area over an additional winter season Egli et al. 4 (2012) calculated SCD curves that assumed correlations between HS and the change in HS 5 (dHS), however these curves deviated substantially from observations, suggesting that the main deposition patterns remained during melt and that the statistical properties of SWE 6 7 distribution control SCD. This is parameterised as the SCD curve in many hydrological and 8 land surface models (Kuchment and Gelfan, 1996; Verseghy, 2000; Essery and Pomeroy, 9 2004) where a spatially uniform ablation rate is applied to a frequency distribution of SWE to calculate areal ablation rates and SCD during melt. such correlations did not exist. Neither 10 11 study examined why such correlations were absent.

HoweverOn the other side, spatially varying melt rates – caused by differences in insolation due to aspect (Marks and Dozier, 1992), net solar irradiance due to albedo 13 14 differences (Skiles et al., 2015), internal energy storage due to deep, cold snow (DeBeer and 15 Pomeroy, 2010), turbulent transfer (Pohl et al., 2006) and advected energy due to bare ground 16 or exposed vegetation (Mott et al., 2011; 2013; Ménard et al., 2014) can alter this pre-melt 17 SWE distribution and, when correlated to SWE, result in a spatial variability to SCD (Faria et 18 al., 2000; Pomeroy et al., 20012004; Essery and Pomeroy, 2004; Dornes et al., 2008a, b; 19 DeBeer and Pomeroy, 2010). For instance in the boreal forest, snow interception by 20 needleleaf trees reduces SWE accumulation, but higher ablation rates near tree trunks 21 accelerate melt this induces a negative spatial covariance between premelt SWE and melt 22 rates that causes a strong bias in SCD (Faria et al., 2000). Pomeroy et al. (2003) took 23 measurements of energy fluxes to snowpacks using eddy correlation and slope based 24 radiometers and snow ablation using spatially distributed snow surveys in a Yukon mountain 25 valley in April and found that whilst ablation was proceeding rapidly on south facing slopes 26 where snow was initially shallow, snow accumulation was still occurring on north facing 27 slopes where a large drift had formed. The different snow surface states and impact of 28 prevailing winds and slope and aspect resulted in energy fluxes melting snow on south facing 29 slopes and cooling snow on north facing slopes. As a consequence, a common SCD was not 30 viable over the whole valley. In Yukon and the Canadian Rockies, subsequent studies found melt variations to be important in controlling snow ablation and SCD (Pomeroy et al., 2003; 31 32 2004; Dornes et al., 2008a, b; DeBeer and Pomeroy, 2009; 2010). Pomeroy et al. (2004) 33 reported that different spatial scales and landscape classes influence melt rates to be positively or negatively correlated to pre-melt SWE throughout melt in a wide variety of cold regions 1 2 mountain environments. Dornes et al. (2008a, b) found that hydrological models and land 3 surface schemes that did not consider slope and aspect impacts on melt as well as initial SWE 4 could not be calibrated to produce realistic SCD curves or streamflow discharge hydrographs. 5 Most interestingly, DeBeer and Pomeroy (2010) found in a windswept alpine catchment subject to substantial snow redistribution, that melt rate variations were quite-important in the 6 7 Canadian Rockies induring early melt. In contrast, Winstral and Marks (2014) found 8 modelled SWE and -Grünewald et al. (2010) observed only a weak relationship between 9 topographic and meteorological variables to spatial melt rates in a Swiss mountain valley; the 10 relationship decreased later in the were correlated with r = -0.66 in the mountains of southern 11 Idaho. Such a large correlation between modelled melt season. They found using Terrestrial Laser Scanning (TLS) and SWE would indicate that the variability of pre-melt SWE was 12 13 much more variable than spatial melt differences. Winstral and Marks (2002), Magnusson et al. (2011) and Winstral et al. (2013) concluded that spatial melt models which do not include 14 spatial SWE variations caused by wind effects were not sufficient are relevant to model runoff 15 16 in mountains. Anderton et al. (2004) found that pre-melt SWE was more important than spatial melt differences for explaining SCD correctly in the Pyreneessome regions. 17

When focusing on studies in alpine terrain without a larger vegetation effect on melt 18 19 (e.g. DeBeer and Pomeroy, 2010; Egli et al. 2012) it still remains unclear whether spatial variable snowmelt in addition to spatially variable SWE should be considered in calculating 20 SCD. Much of this uncertainty is due to the limited number of detailed measurements 21 available and the uncertainty of distributed model results. There are likely to be fundamental 22 23 differences in the sequence of ablation between relatively warm and cold mountain environments due to the effect of internal energy deficits in delaying melt of deep snowpacks 24 25 (DeBeer and Pomeroy, 2010). But there are also substantial differences in solar irradiance as the summer solstice is approached as shown in Fig. 1 in April at  $50^{\circ}$  N there is a 17 MJ m2 26 27 difference between irradiance on 30% N and S facing slopes, but by solstice this decreases to 28 2.5 MJ m2. The variation in observations and model results for alpine terrain creates 29 uncertainty in the relative importance of spatial melt rates and SWE to determine SCD and areal ablation rates. To address this uncertainty an extremely high spatial resolution digital 30 surface model dataset was collected over an alpine ridge in the Canadian Rockies using an 31 Unmanned Aerial Vehicle (UAV) and Structure from Motion (SfM) analysis to repeatedly 32 determine snow depth and snow depth differences during a melt season as proxies of initial 33

SWE and sequential, spatial melt rates. Analysis over several spatial scales was conducted to correlate variables that might influence melt to measured spatial melt rates and investigate the influence of initial SWE and spatial variation in melt on SCD and areal mean melt over an alpine ridge in high mountain terrain.

Regional differences, e.g. the complexity of the terrain and wind redistribution, will alter the dominance of SWE variability on SCD and may thus explain part of the different findings of various studies. Not all cited studies fall in the highest CV category suggested by Liston et al. (2004). Furthermore, in study regions with a large elevation gradients, altitudinal melt energy differences as well as precipitation phase differences will play an important role in governing SCD (Blöschl and Kirnbauer, 1992; Elder et al., 1998).

From a practical modelling perspective, it is simpler to explicitly calculate melt energy differences in a model (Marks and Dozier, 1992) than to calculate snow redistribution mechanistically over complex terrain (Liston and Sturm, 1998; Mott et al., 2010; Fang et al., 2013; Musselman et al., 2015). Empirical modelling of SWE variability (Luce et al., 1999; Winstral and Marks, 2002; Liston et al., 2004, Essery and Pomeroy, 2004; Helbig et al., 2015) has therefore been a preferred choice. However, Dornes et al. (2008a, b) showed substantial interannual variability of SWE distributions and correlations to melt rate which invalidated empirical assumptions in some years and caused predictive failure of land surface hydrology models. Without explicit information on the effects of redistribution on the distribution of SWE, it is not possible to estimate the impact of correlations between SWE and melt rate on SCD curves.

This study aims to show the influence of peak SWE variability and melt rate variability and 
[revised manuscript text omitted]
 snowissnow is possible, although the snowpack is was 6 7 quite ripe at this time of the year.

In the first period, P1, *Brightness* had a large effect in the northern subarea (r = -0.66). 9 Figure 5a visualizes this relationship between dark snow and meltdHS. The high scatter 10 especially for brighter snow pixels can partly be explained with radiation differences. For the 11 same period and area *Solar Irradiance* and *Deviation from North* had a correlation of r = 0.57. 12 Figure 5b illustrates the dependency with Solar Irradiance but for white pixels only (approximately 50% of the observations). A clear dependency is visible with a correlation 13 14 coefficient of r = 0.66. Radiation effects were more substantial during P2 in this northern 15 subarea with r = 0.84 for both *Solar Irradiance* and *Deviation from North*. This may be 16 explained due to less scatter produced by albedo differences in this period (r = 0.03). Darker 17 parts of the snowcover melted out by the end of this period.

The correlations of dHS with Brightness, Deviation from North and Solar Irradiance were often strong. dHS increased from 5 to 7 cm/d (nearly 60% increase) as aspect shifted about 115 deg from north to south or snow from clean to dusty (c.f. Fig. 55b). This shows the importance of spatial variation in net solar irradiance to melt energetics – as exemplified by the modelmodelled energy budget shown in Fig. 4b.- The impact of dust on albedo and slope on solar irradiance is well established in the snow literature and so this is expected. What is a more unique finding here is that dHS is not correlated largely with initial SWE (HS0, Table 1) as found by DeBeer and Pomeroy (2009, 2010), Pomeroy et al. (2003, 2004), Dornes et al. (2008 a, b) and other mountain studies in Canada. This indicates a lack of covariance between melt rate and SWE in late melt that should have important implications for SCD curves (Pomeroy et al., 2001, section 3.4.3).

All previous studies in the Canadian Rockies, Alberta and Coast Mountains, Yukon focussed on the full melt period rather than the late melt period that is measured in this study and so the importance of season differences in irradiance to slopes as shown in Fig. 1 and the late melt isothermal snowpacks may be important to explaining the missing spatial correlation between melt and SWE. During early melt the cold content is related to snow depth, which 1 2 likely will result in a spatial correlation between SWE and melt (c.f. DeBeer and Pomeroy, 2010, 2017). Furthermore, the observed two dominant wind directions related to precipitation 3 and strong wind speeds have influenced spatial SWE patterns and reduced the likelihood of a 4 5 spatial correlation of SWE and melt. In contrast, areas with wind transport from primarily one direction and hence preferential SWE loading onto slopes will affect particular aspects, which 6 7 in turn may be southerly, and hence induce a spatial correlation between SWE and ablation. 8 Another reason for a missing spatial correlation is discussed in section What is a more 9 interesting finding here is that dHS was not correlated with initial HS0, Fig. 5c, Table 1), as 10 was observed in other cold regions mountain studies in Canada such as DeBeer and Pomeroy 11 (2009, 2010), Pomeroy et al. (2003, 2004), and Dornes et al. (2008 a, b). A lack of covariance between HSO and dHS in late melt has important implications for SCD curves (Pomerov et 12 13 al., 2001), which will be highlighted in section 3.5. Figure 5c shows the areal mean values for 14 HS0 and dHS for flat areas (slope  $< 5^{\circ}$ ) and areas on both sides of the ridge (threshold aspect 15 is 235°, slope  $\geq$  5°). The hypothesis for this study period was that large drifts on south-facing parts of the ridge cause a correlation between melt energy and SWE. Indeed, the southeast 16 17 part showed larger HSO and dHS compared to the flat and northwest part of the study area. 18 This suggests a correlation between HS0 and dHS, which was not apparent when analysing all pixels. In each subarea the range of snow depth was large, which diminished the observed 19 correlation. More importantly, on the south-eastern face a mild negative correlation of r = -20 21 0.35 developed (Fig. 5d), which may be explained by a remaining cold content in deep drifts. 22 This negative correlation is not apparent for smaller dHS values, in the northwest part of the 23 ridge (Fig. 5e). The lack of correlation in the Fig. 5c point cloud was contributed to by 24 compensation between the positive correlation driven by melt energy and the negative 25 correlation from a cold content.

To aid in analysing the reasons for the lack of correlations between Hs and dHS in this study area one can formulate some prerequisites for large spatial correlations in general. For instance, cold content has the potential to establish a negative correlation since deeper snowpacks take longer to warm up to 0 °C and so shallower snowpacks start melting earlier. This results in greater melt for shallower snowpacks. The spatial distribution of SWE and melt energy on slopes may result in a negative or positive correlations, which depend on whether deep drifts are found on north-facing or south-facing slopes. For a large correlation between Hs and dHS, either snow redistribution to slopes or deep snow cold content

processes needs to be present and need to not counteract each other. In such a case the sign of 1 2 the correlation driven by spatial distribution of SWE melt energy must be negative (drifts on 3 north-facing slopes) and hence similar to the negative correlation driven by greater cold 4 content in deeper snow. Remote sensing techniques such as remote sensing can determine 5 where deep drifts occur on north-facing slopes (Wayand et al., 2018; Painter et al., 2016) and these are quite prevalent in many regions. DeBeer and Pomeroy (2010) showed that spatial 6 7 variation in cold content was large only in early melt and was unimportant to SCD later in the 8 melt season when isothermal snowpacks predominate.

Given these scenarios some guidelines for modelling areal SCD can be provided. Models must be able to represent realistic correlations between SWE and melt in order to model the effect of this correlation on SCD (Essery and Pomeroy, 2004). Potential pitfalls are incomplete modelling representations that might neglect a governing process. To capture the spatial correlations, models need to include snow redistribution, internal snowpack energetics and melt rate variability on slopes at fairly fine scales (<100 m) in complex terrain. Semidistributed models with homogenous snow distribution over large areas or distributed models that neglect blowing snow redistribution may misrepresent spatial correlations of SWE and melt.

Another reason for models misrepresenting spatial correlations between HS0 and dHS is discussed in section 3.63.56, in which the mismatch of scales of ablationdHS and SWEHS0 patterns areis discussed.

The stepwise linear regression results shown in Table 2 confirm that the most important variables explaining ablation variation are solar irradiance and albedo. Combinations of solar irradiance and albedo increased the explanation compared to univariate regressions (Table 1). For example for P1 in the northern subarea, a model with *Deviation from North* and *Brightness* explained nearly 70% of the total ablation variance with nearly equally large (normalized) coefficients, indicating equal effect contributions of irradiance and albedo to explaining variations in dHS.

**3.4 Relative importance of ablation and initial SWE**

**3.4.13.4 Variability of ablationdHS in relation to (initial) SWEHS0 and temporal persistence**

Table 32 shows mean, standard deviation and CV values of HS and dHS in different periods and subareas. Throughout the melt season CV values of dHS were about five times smaller than those of HS. At the start of the study period, the variability of dHS was smaller than that of HS by a factor 3.7 to 6.7, for the whole area approximately by a factor 5. Applying the mean measured snow density from 19 measurements between 19 May and 22 May (413 kg/m3) to HS provides an estimate of mean initial SWE of 520 mm with a standard deviation of over 480 mm. Ablation amounts in period P1 were in mean 334 mm, with a standard deviation of only 65 mm. These values indicate much larger SWE variability than ablation variability in this period.

**13 3.4.2 Persistence of ablation patterns**

For the whole area only a weak correlation (r = 0.36) was found between ablation 15 patterns betweenover the two long periods P1 and P2. Larger correlations were found for the northern subarea (r = 0.60). Ablation patterns in certain sub-periods with similar weather 16 17 conditions were correlated to each other. For instance, ablation patterns in the cool and 18 cloudy period between 05 May and 01 June were correlated with two other rather cloudy sub-19 periods at the end of the study period with r = 0.49 and r = 0.64, and to the later combined 20 period P2 (r = 0.70). Closer investigations Further investigation on how these correlations 21 responded to weather conditions werewas not possible given the reduced signal-to-noise ratio 22 for shorter time periods. Larger periods always included and the inclusion of several types of 23 weather patternstypes over longer periods.

**24 3.4.33.5**

**B.5** Depletion curves**

Maximum differences in melt ratedHS of up to 100% were measured (section 3.3) and melt rates were spatially persistent especially in the northern subarea. Similarly to Pomeroy et al. (2001) and Egli et al. (2012) the impact of spatial melt ratesdHS on snow-cover depletion and areal melt were analysed in several scenarios:

Variable HS0/uniform meltdHS: This scenario started with the measured distribution of HS at the start of the study period (HS0) and a spatially uniform melt ratedHS value was applied for each pixel. This melt ratevalue was determined with observed mean ablation values shown in Table 3. Each pixel was reduced by this mean melt ratevalue and any negative values in HS were set to 0. SCA was defined as the ratio of the number of grid points with HS > 0 to all pixels.

- 2. Uniform HS0/variable meltdHS: In this scenario, the mean initial snow depth as shown in Table 3 was uniformly distributed in the whole snow-covered area. Spatially variable melt ratesdHS values as measured with the UAV were applied to each pixel. To obtain the exact melt-\_out time this scenario was calculated in a daily resolution with using a temporally constant melt ratedHS value between flights. No exact meltdHS amounts arewere available for pixels which have melted out between flights. For those pixels the mean melt rate were areal dHS value was applied. The general shape of SCD curves can be obtained when this scenario iswas also calculated on the time resolution of the UAV flights.
  - Uniform HS0/uniform meltdHS: Similar to scenario 2, but a spatially uniform melt ratedHS value was applied to each pixel, each of which had a uniform HS0. This scenario was also calculated on a daily resolution.

In all scenarios, SCA was set to 1 for the area which was snow-covered at the start of the study period. Figure 6 shows mean HS ablation and SCD curves for the whole area and the northern subarea (top), for which more flights are available. Differences between measured development and the first scenario of uniform meltdHS and variable HSO were not large. However, a large difference between measurements and the second and third scenario of scenarios with uniform HS0 withand either variable or uniform meltdHS is obvious. Areal meltdHS in those scenarios was overestimated before modelled melt--out because of overestimatingthe overestimation of SCA. Later during melt, areal meltdHS was underestimated (or zero) since nearly (most or all) snow disappeared too early. This is particularly important when the aim is to model late rain-on-snow events in hydrological models. For this area these (Pomeroy et al., 2016). These results indicate that it is possible ignoring to not represent the spatial melt variability in late melt to achieve and still simulate a realistic SCD curve, while this is not possible ignoringif the spatial HSO-variability of HSO is not represented. This main feature is consistent with Egli et al. (2012).

The main reason why the observed dHS differences, which were substantial and partly persistentmelt differences were, did not largely influencing depletioninfluence SCD curves compared to a homogeneous melt scenario can be found in the small or missingto negligible spatial correlation between meltdHS and initial SWEHS0 (cf. section 3.3 and Table 1). Large correlations substantially influence SCD: Negative correlation accelerates SCD at the beginning of melt and delays it in late melt lengthening the snowmelt season and vice versa with positive correlations (Essery and Pomeroy, 2004).

In case of weak to noWhere correlation is insignificant, spatial melt differences can be quite large without affecting SCD curves-compared to homogeneous melt. The. In this case, spatially variable melt can be viewed as a nearly random process. There is still some impact because of the – it introduces noise into the log-normal frequency distribution of SWE.HS, but does not affect the emergent behaviour of the SCD curve. Here, with a much larger variability of SWEHS0 compared to meltdHS (see section 3.4.1)3.4) and only small spatial correlations between them (see Table 1), SWE must dominateHS0 controls the SCD.

**15 3.53.6 Scale dependencies of meltdHS**

Figure 7 and 8 show how the variance of dHS, the variance of explaining variables and correlations thereof, develop with larger lag distance between point pairs (variograms and correlograms, Eqs. 1 to 3). This gives further insights into the driving factors of ablation and why a correlation between dHS and initial HSO was weak in this study area during late melt.

In Fig.Figure 7a shows with, the variogram of dHS, shows that the variance increased over two distinct length scales, one less than 50 m and one greater than 200 m. This implies that the driving forces toprocesses which generate variance for ablationdHS need to be searchedinvestigated at these two scales. In section 3.3 a strong correlation was found between dHS and *Brightness* and *Solar Irradiance*, but only small correlations tobetween these and HS0. These variables were alsotherefore analysed with variograms and correlograms.

The variogram of *Brightness* shown in Fig. 7b indicates a variance increase only at the small lag distances less than 50 m. This is consistent with the visual impression of a smallscale variability of albedo shown in Fig. 3b. The correlogram shown in Fig. 7c reveals a strong correlation of *Brightness* with dHS at these small scales ( $\rho_{xy} \approx -0.6$  at 50 m lag distance). This demonstrates that albedo was largely responsible for the small-scale meltdHS
 variability observed in Fig. 7a.

Figure 8a shows the variogram of *Solar Irradiance*. A small increase for length scales less than 100 m suggests radiation and aspect differences at those scales (within-slope variations), but the largest increase can be observed at lag distances longer than 200 m. This scale represents slopes on both sides of the ridge and coincides with the larger scale of meltdHS variance. Indeed, the correlogram (Fig. 8b) confirms that the largest correlation with meltdHS to  $\rho_{xy} = 0.4$  was achieved at those larger distances.

The same analysis for initial snow depth (HS0) can be seen in Fig. 8c and d. Most of 10 the variance for snow depth is at length scales less than 100 m. The periodic behaviour shown 11 beyond that scale may be due to the patchy snow cover which has long snow-free patches. No 12 largesubstantive correlation with dHS is observable on all scales (Fig. 8d).

This analysis offers anfurther explanation why ablationdHS and initial SWEHSO were not spatially correlated in these observations. MeltdHS variance (ignoring the small scale influence of albedo) was related to –large scale aspect changes on both slopes, while and medium scale albedo change, whilst snow depth was mainly variable mainly at much smaller scales. This scale mismatch prevented a stronger correlation.

**18 **3.6 Comparison with other studies in alpine terrain**

The correlation coefficients leads to a larger scatter between dHS and explaining 20 variables found here are larger than those found by Grünewald et al. (2010) in a Swiss alpine catchment. They found maximum correlations of  $|\mathbf{r}| \approx 0.4$  for altitude, slope, northing and 21 22 initial SWE, mainly for the first of their ablation periods. This is despite the fact that they 23 used a wider range of explaining variables such as wind fields from a high resolution wind 24 flow model and accounted for time-variant diffuse radiation in modelling shortwave irradiance. The lower relation to explaining variables may be caused by regional differences, 25 but can also be found in the slightly larger area (0.6 km2) studied by Grünewald et al. (2010), 26 which can potentially include more effects. These local effects were observable in our study 27 as correlations change strength and sign with time and space (c.f. Table 1). Grünewald et al. 28 29 (2010) found correlations diminished with time and explained this by suggesting the increasing importance of local advection of heat. These diminishing correlations were not 30 31 observed here. They also found more similar variance in ablation and SWE. This may be explained by the particularly wind swept study site. HS0 values and thus prevented a substantive spatial correlation.

Egli et al. (2012) working in the same catchment as Grünewald et al. (2010) found that SCD curves were insensitive to the degree of heterogeneity of ablation. This can be explained as they also did not find a large correlation between SWE and melt. As also found here, Egli et al. (2012) observed that when correlations developed, they were temporally and spatially unsteady, disappeared or changed sign. This reduced the impact of these small-scale correlations on SCD over the ablation season in within a larger study area.

In a northern Canadian mountain basin (Yukon), Pomerov et al. (2004) observed on their smallest scale (100 to 300 m) a large negative correlation between ablation and SWE of r = -0.95 at the valley bottom part of their 660 m long transect, a correlation of r = -0.63 on the south face and no correlation on the north face slopes (c.f. their Fig. 5). This is in agreement to findings here that correlations vary regionally. Pomeroy et al. (2004) explained those differences with by varying exposures of vegetation, which is not a factor in this study. When these three slopes are aggregated to the sub-basin, the areal multi-scale correlation is diminished (Fig. 5, Pomeroy et al., 2004). The large correlation of r = -0.86 over the sub-basin is driven by a slope scale association between snow redistribution to north faces that also experience lower ablation rates (Fig. 6, Pomeroy et al., 2004). This is a meso-scale feature of southerly winds in the basin due to proximity of the Pacific Ocean to the south. Dornes et al. (2008) showed that representing differences in ablation rates amongst these slopes is critical to calculating accurate SCD, but did not suggest that small-scale ablation rate variation need to be considered. They found that by disaggregating the basin into slope units with averaged melt energy applied to each unit, then accurate SCD curves could be estimated within each slope unit using the variability of SWE alone.

DeBeer and Pomeroy (2010; 2017) concluded that multi-scale variable melt and SWE improved SCD modelling compared to aerial photography of SCA during early melt, but not mid-or-late-melt seasons in the Canadian Rockies (Marmot Creek Basin). They included a spatial distribution of SWE within four slope-scale subareas, and modelled the cold content of snow, which introduced an early multi-scale correlation between SWE and melt in the model. Applying different melt rates within each of the slope-based subareas improved simulations of SCD compared to uniform melt rates during early melt. Considering the whole ablation season they concluded that "...the improvements from including simulations of inhomogeneous melt over the entire snowmelt period in the spring were negligible (Table 3)." This refers to small scale inhomogeneous melt and is in agreement with the measurements presented here. Over Fisera Ridge in the same region, Musselman et al. (2015) showed a slope scale but not fine scale spatial association between ablation and SWE, and noted that the slope scale association was due to the localized wind loading of this particular ridge (northerly winds) and would not apply to the larger basin studied by DeBeer and Pomeroy (2017) or Pomeroy et al. (2016) where wind directions varied.

2

7

Winstral and Marks (2014) found modelled SWE and melt rates were correlated with r = 0.66. This large modelled correlation of spatial patterns of SWE and melt may be realistic in this study area, since snow transport is dominated there by a rather homogeneous wind direction, both in space and time which leads to a coincidence between preferentially loaded slopes and melt energy. Such a large correlation between modelled melt and SWE would indicate that spatial melt differences are relevant to model SCD correctly in this area.

In summary, some studies found correlations between melt and SWE at slope scales, but not at fine scales. These associations were strongest early in melt and at higher latitudes and where wind redistribution occurred over consistent directions due to synoptic conditions during mesoscale wind loading of slopes and is consistent with the slope based solar irradiance differences shown in Fig. 1. However, no study showed consistent and persistent fine scale association between ablation and SWE suggested that they can be considered uncorrelated in modelling at fine scales. To address differences in melt energy at slope scales, modellers can chose to calculate averaged energetics to slope units and apply a mean ablation rate to a frequency distribution of SWE over the slope as was demonstrated by Dornes et al. (2008) and DeBeer and Pomeroy (2010; 2017). This is computationally more efficient than the fully distributed calculations employed by Winstral and Marks (2014) and Musselman et al. (2015) and is a promising and likely necessary direction for disaggregation of land surface schemes calculations of melt in mountain regions.

Two processes were previously discussed and described in Fig. 5c which could drive compensating correlations between HSO and dHS; cold content and melt energy. Cold content likely acts on a similar scale as HSO, since it depends mainly on snow depth. As shown in Fig. 5d and 5e a negative correlation driven by cold content is not uniformly present. Melt energy differences, i.e. differences in net shortwave radiation, turbulent fluxes, and net longwave radiation, are not directly dependent on snow depth, but need to spatially coincide by chance 1 2 (e.g. by direction of redistribution). Acknowledging that Solar Irradiance is a simple proxy of melt energy, spatial coincidences between accumulation and melt energy are only present 3 over larger distances (Fig 8b). The large scatter between HS0 and dHS results from the 4 5 observation that most of the variance of HSO occurs at much smaller scales (Fig. 7a). Figure 8d illustrates variability in the compensating correlations. At small scales below 50 m, the 6 7 differences in *Solar Irradiance* are small and the cold content is responsible for slight 8 negative correlation between HSO and dHS. This is counteracted by Solar Irradiance until the 9 distance of 250 m (cp. Fig 8a).

There needs to be a match in scaling behaviour between SWE and melt rate for these variables to develop spatial correlations. Assuming melt is primarily driven by aspect and 11 12 slope differences as in the proxy Solar Irradiance, SWE must vary on similar scales for a correlation to develop. This may be achieved if SWE varies primarily over larger scales, e.g. 13 14 in a simple topography of a ridge without gullies and with one predominant wind direction 15 during blowing snow, in which one slope face has much larger SWE values than the other. This may also be achieved if Solar Irradiance acts on a smaller scale similar to HS0. This 16 might be possible in highly complex terrain in which most slope/aspects differences can be 17 18 found on scales below 100 m but this does not correspond to the "ridge" in our study site.

**4 Conclusions and outlook**

The aim of this study was to determine factors which influence areal snow ablation 21 patterns in alpine terrain-using spatially intensive observation. The dependency of SWEsnow 22 accumulation and topographic variables on with spatial melt rates were analysed for an alpine 23 ridge in the Fortress Mountain Snow Laboratory located in the Canadian Rocky Mountains. Detailed maps of snow depth (HS), snow depth changes (dHS)change and snow-covered area 24 25 (SCA)—were generated during late season ablation with UAV—based orthophotos, photogrammetry and Structure-from-Motion techniques. HSSnow depth and dHSits change 26 27 served as proxies for SWEsnow accumulation and melt rates. Ablation rates Snow depth 28 change values were found to be spatially variable and mainly dependent on variation in solar irradiance and albedo-, and likely on the cold content of the snowpack which is a function of 29 snow depth. Local and small-scale dust variations, which have neverhad not previously been 30 observable to this degreeobserved in the area, increased the variability of ablation. 31

However, snow-cover Snowcover depletion (SCD) curves were largely mostly 2 dominated by the SWE variability of initial snow depth at the start of this study, which rather 3 than the variability in snow depth change. Initial snow depth variability was approximately 4 five times larger than melt the variability in snow depth change in this extraordinarily 5 windswept environment. More importantly, SWE and melt rates The scales of variability of snow depth and snow depth change were mismatched, with snow depth variability occurring 6 7 at small scales (<10 m) and snow depth change associated with the medium scale (50 m) of 8 albedo variation or the slope scale (100s of m) of solar irradiance variation. As a result, the 9 initial snow depth and changes in snow depth were not strongly correlated over space, which 10 is a prerequisite for melt influencing SCD. Three reasons for lack of spatial association 11 between ablation and SWE patterns here are: (1) the snowcover was isothermal during of the 12 study period so that spatial differences in the depth dependent cold content as found by 13 DeBeer and Pomeroy (2010), did not play a relevant role; (2) the SWE pattern was and so only initial snow depth influenced by two dominating wind directions, preventing wind loading on 14 particular slopes coincident with either larger or low energy input as found by Pomeroy et al. 15 16 (2003), Dornes et al. (2008 a,b) and Musselman et al. (2015); (3) near summer solstice conditions limited differences of radiation energy input between slopes; (4) a scale mismatch 17 18 between the variabilities of ablation and SWE was detected, with SWE varying mostly on 19 smaller scales (in slope gullies, ridges), while melt varied mostly on larger slope scale aspect differences. snowcover depletion. 20

These findings suggest that during those conditions SCD curves can be calculated without the spatial distribution of melt rates, while hydrological and atmospheric models need to implement a realistic distribution of SWE in order to do this. Comparison of these results to those found in Switzerland, Yukon, the Canadian Rockies and mountains in Idaho, indicates that clear advice to modellers is still not possible. It is not possible to determine without detailed modelling or measurements whether, when and where a catchment wide multi-scale association between SWE and melt are capable to sufficiently alter SCD curves from those derived with an uniform melt assumption.

Thus The observations collected here show the prerequisites for strong correlations that can impact snowcover depletion curves. Correlation between melt and snow accumulation may be driven by cold content and melt energy distributions. Whilst cold content can create a negative correlation between melt and snow accumulation, melt energy variations can create either positive or negative correlations. In order to not compensate for each other, one process
 needs to be dominant, or the both processes need to create a similar negative correlations. It
 is also important that these variations occur at the same spatial scales.

To further investigate these arguments, longer time series of spatially detailed 5 SWEsnowpack and snowcover observations need to be made in order to further test and 6 examine the temporal evolution of the spatial covariance and variance of ablation and SWE in 7 order to accumulation in various global alpine environments. The results of such a study 8 could suggest how to efficiently and accurately model parameterise snow-cover depletion and 9 runoff in snow-meltmodels for snowmelt dominated catchment, and to deal with region 10 variations in associations between SWE and meltalpine catchments, without relying on 11 powerfulmodel calibration-routines. This will help to transfer snow-hydrological models to 12 ungauged catchments and to model future climate scenarios where snow redistribution patterns might be vastly different. 13

**15 **5 Data availability**

The data is available, upon request from the database manager (Branko 17 Zdravkovic), Amber Peterson, in the Changing Cold Regions (CCRN) Global Water Futures 18 dataserver. (www.ccrnetwork.ca/outputs/data/index.php). Please refer to this website for 19 contact details. The data involves all UAV derived grids for HS, dHS and SCA, as well as 20 grids of explaining variables (Brightness, Deviation from North and Slope) in 1 m resolution 21 (cp. section 2.4). Metadata is provided which explains the file naming convention of the grids 22 (dates and variables).

**24 6 Acknowledgement**

The authors wish to acknowledge Phillip Harder for UAV training, Chris Marsh for post-processing and modelling support, May Guan and Angus Duncan for extensive field work help, Xing Fang for the assistance in modelling CHRM results, as well as Phillip Harder, Jonathan Conway, Keith Musselman, Nico Leroux and Nik Aksamit for and helpful comments and discussions. We are grateful for logistical support from Fortress Mountain Ski Resort, Cherie Westbrook for access to a differential GPS unit. Funding was provided by NSERC through Discovery and Research Tools and Instruments grants and NSERC's Changing Cold Regions Network, by-the Canada Research Chairs and Canada Excellence
 Research Chairs programmes, Alberta Innovation, Global Water Futures, and by Alberta
 Agriculture and Forestry.

**4 7 References**

- Anderton, S. P., White, S. M., & Alvera, B. (2004). Evaluation of spatial variability in
  snow water equivalent for a high mountain catchment. Hydrological Processes, 18(3), 435453.
- 8 Blöschl, G. & Kirnbauer, R. (1992). An analysis of snow cover patterns in a small
alpine catchment. Hydrological Processes, 6(1), 99-109.

Clark, M.P., Hendrikx, J., Slater, A.G., Kavetski, D., Anderson, B., Cullen, N.J., Kerr,
 T., Hreinsson, E.Ö. & Woods, R.A. (2011). Representing spatial variability of snow water
 equivalent in hydrologic and land-surface models: A review. Water Resources Research,
 47(7).

**DeBeer, C. M., & Pomeroy, J. W. (2009). Modelling snow melt and snowcover depletion in a small alpine cirque, Canadian Rocky Mountains. Hydrological processes, 23(18), 2584-2599.**

DeBeer, C. M., & Pomeroy, J. W. (2010). Simulation of the snowmelt runoff 18 contributing area in a small alpine basin. Hydrology and Earth System Sciences, 14(7), 1205.

DeBeer, C. M., & Pomeroy, J. W. (2017). Influence of snowpack and melt energy
heterogeneity on snow cover depletion and snowmelt runoff simulation in a cold mountain
environment. Journal of Hydrology, 553, 199–213.

Deems, J. S., Fassnacht, S. R., & Elder, K. J. (2006). Fractal distribution of snow
depth from LiDAR data. Journal of Hydrometeorology, 7(2), 285-297.

Donald, J. R., Soulis, E. D., Kouwen, N. & Pietroniro, A. (1995). A Land Cover25 Based Snow Cover Representation for Distributed Hydrologic Models. Water Resources
Research, 31, 995-1009.

Dornes, P. F., Pomeroy, J. W., Pietroniro, A., Carey, S. K. & Quinton, W. L. (2008a).
Influence of landscape aggregation in modelling snow-cover ablation and snowmelt runoff in
a sub-arctic mountainous environment. Hydrological Sciences Journal, 53, 725-740.

| 1  | Dornes, P. F., Pomeroy, J. W., Pietroniro, A. and Verseghy, D. L. (2008b): Effects of         |
|----|-----------------------------------------------------------------------------------------------|
| 2  | spatial aggregation of initial conditions and forcing data on modeling snowmelt using a land  |
| 3  | surface scheme. Journal of Hydrometeorology, 9, 789-803.                                      |
| 4  | Egli, L., Jonas, T., Grünewald, T., Schirmer, M., & Burlando, P. (2012). Dynamics of          |
| 5  | snow ablation in a small Alpine catchment observed by repeated terrestrial laser scans.       |
| 6  | Hydrological Processes, 26(10), 1574-1585.                                                    |
| 7  | Elder, K., Rosenthal, W., & Davis, R. E. (1998). Estimating the spatial distribution of       |
| 8  | snow water equivalence in a montane watershed. Hydrological Processes, 12(10-11), 1793-       |
| 9  | 1808.                                                                                  |
| 10 | Essery, R., & Pomeroy, J. (2004). Implications of spatial distributions of snow mass          |
| 11 | and melt rate for snow-cover depletion: theoretical considerations. Annals of Glaciology,     |
| 12 | 38(1), 261-265.                                                                               |
| 13 | Fang, X., Pomeroy, J. W., Ellis, C. R., MacDonald, M. K., DeBeer, C. M., & Brown,             |
| 14 | T. (2013). Multi-variable evaluation of hydrological model predictions for a headwater basin  |
| 15 | in the Canadian Rocky Mountains. Hydrology and Earth System Sciences, 17(4), 1635-1659.       |
| 16 | Faria, D. A., Pomeroy, J. W. and Essery, R. L. H. (2000): Effect of covariance                |
| 17 | between ablation and snow water equivalent on depletion of snow covered area in a forest.     |
| 18 | Hydrological Processes, 14, 2683-2695.                                                        |
| 19 | Gray, D. M., & Male, D. H. (Eds.). (1981). Handbook of snow: principles, processes,           |
| 20 | management & use. Pergamon.                                                                   |
| 21 | Grünewald, T., Schirmer, M., Mott, R., & Lehning, M. (2010). Spatial and temporal             |
| 22 | variability of snow depth and ablation rates in a small mountain catchment. Cryosphere, 4(2), |
| 23 | 215-225.                                                                                      |
| 24 | Harder, P., Schirmer, M., Pomeroy, J. & Helgason, W. (2016). Accuracy of snow                 |
| 25 | depth estimation in mountain and prairie environments by an unmanned aerial vehicle. The      |
| 26 | Cryosphere, 10, 2559-2571.                                                                    |
| 27 | Kuchment, L. & Gelfan, A. (1996). The determination of the snowmelt rate and the              |
| 28 | meltwater outflow from a snowpack for modelling river runoff generation. Journal of           |
| 29 | <del>Hydrology, 179, 23 - 36.</del>                                                           |
|    |                                                                                               |

Helbig, N., van Herwijnen, A., Magnusson, J., & Jonas, T. (2015). Fractional snow-2 covered area parameterization over complex topography. Hydrology and Earth System Sciences, 19(3), 1339-1351. 3 4 Liston, G. E., & Sturm, M. (1998). A snow-transport model for complex terrain. 5 Journal of Glaciology, 44(148), 498-516. 6 Liston, G.E. (1999). Interrelationships among snow distribution, snowmelt and snow 7 cover depletion: implications for atmospheric, hydrologic and ecologic modeling. Journal of 8 Applied Meteorology, 38, 1474-1487. 9 Liston, G.E. (2004). Representing subgrid snow cover heterogeneities in regional and global models. Journal of Climate, 17, 1381-1397. 10 Luce, C. H., Tarboton, D. G., and & Cooley, K. R. (1998). The influence of the spatial 11 12 distribution of snow on basin-averaged snowmelt. Hydrological Processes, 12, 1671-1683. 13 Luce, C. H., Tarboton, D. G., and & Cooley, K. R. (1999): Sub-grid parameterization 14 of snow distribution for an energy and mass balance snow cover model. Hydrological 15 Processes, 13, 1921–1933. 16 MacDonald, M. K., Pomeroy, J. W. & Pietroniro, A. (2010). On the importance of 17 sublimation to an alpine snow mass balance in the Canadian Rocky Mountains. Hydrology 18 and Earth System Sciences, 14, 1401-1415. 19 Magnusson, J., Farinotti, D., Jonas, T., & Bavay, M. (2011). Quantitative evaluation of different hydrological modelling approaches in a partly glacierized Swiss watershed. 20 Hydrological Processes, 25(13), 2071-2084. 21 22 Marks, D., & Dozier, J. (1992). Climate and energy exchange at the snow surface in 23 the alpine region of the Sierra Nevada: 2. Snowcover energy balance, Water Resources 24 Research., 28, 3043-3054. 25 Marsh, P. and Mott, R., Schirmer, M., Bavay, M., Grünewald, T., & Lehning, M. 26 (2010). Understanding snow-transport processes shaping the mountain snow-cover. The 27 Cryosphere, 4(4), 545-559. Pomeroy, J. W. (1996). Meltwater fluxes at an arctic forest-tundra site. Hydrological 28 29 Processes, 10, 1383-1400.

Ménard, C. B., Essery, R. and Pomeroy, J. (2014). Modelled sensitivity of the snow regime to topography, shrub fraction and shrub height. Hydrology and Earth System Sciences, 18, 2375-2392.

Mott, R., Egli, L., Grünewald, T., Dawes, N., Manes, C., Bavay, M., & Lehning, M. (2011). Micrometeorological processes driving snow ablation in an Alpine catchment. The Cryosphere, 5(4), 1083-1098.

Mott, R., Gromke, C., Grünewald, T., & Lehning, M. (2013). Relative importance of
advective heat transport and boundary layer decoupling in the melt dynamics of a patchy
snow cover. Advances in Water Resources, 55, 88-97.

Mott, R., Paterna, E., Horender, S., Crivelli, P. & Lehning, M. (2016). Wind tunnel
experiments: cold-air pooling and atmospheric decoupling above a melting snow patch. The
Cryosphere, 10, 445-458.

Musselman, K. N., Pomeroy, J. W., Essery, R. L. H. & Leroux, N. (2015). Impact of
windflow calculations on simulations of alpine snow accumulation, redistribution and
ablation. Hydrological Processes, 29, 3983-3999.

Painter, T. H., Berisford, D. F., Boardman, J. W., Bormann, K. J., Deems, J. S.,
Gehrke, F., ... & Mattmann, C. (2016). The Airborne Snow Observatory: Fusion of scanning
lidar, imaging spectrometer, and physically-based modeling for mapping snow water
equivalent and snow albedo. Remote Sensing of Environment, 184, 139-152.

Painter, T. H., Deems, J. S., Belnap, J., Hamlet, A. F., Landry, C. C., & Udall, B.
(2010). Response of Colorado River runoff to dust radiative forcing in snow. Proceedings of
the National Academy of Sciences, 107(40), 17125-17130.

Pohl, S., Marsh, P. & Liston, G. (2006). Spatial temporal variability in turbulent fluxes
during spring snowmelt. Arctic, Antarctic, and Alpine Research, 38, 136-146.

Pomeroy, J., & Gray, D. (1995). Snowcover-accumulation, relocation and
management. National Hydrology Research Institute Science Report No. 7. Saskatoon,
Canada.

Pomeroy, J. W., Gray, D. M., Shook, K. R., Toth, B., Essery, R. L. H., Pietroniro, A.
& Hedstrom, N. (1998). An evaluation of snow accumulation and ablation processes for land
surface modelling. Hydrological Processes, 12, 2339-2367.

Pomeroy, J. W., Hanson, S., & Faria, D. (2001). Small-scale variation in snowmelt
 energy in a boreal forest: an additional factor controlling depletion of snow cover. In
 Proceedings of the Eastern Snow Conference, 58, 85-96.

Pomeroy, J. W., Toth, B., Granger, R. J., Hedstrom, N. R., & Essery, R. L. H. (2003).
Variation in surface energetics during snowmelt in a subarctic mountain catchment. Journal of
Hydrometeorology, 4(4), 702-719.

- Pomeroy, J., Essery, R., & Toth, B. (2004). Implications of spatial distributions of
  snow mass and melt rate for snow-cover depletion: observations in a subarctic mountain
  catchment. Annals of Glaciology, 38(1), 195-201.
- Pomeroy, J., Fang, X. and Ellis, C. (2012): Sensitivity of snowmelt hydrology in
   Marmot Creek, Alberta, to forest cover disturbance. Hydrological Processes, 26, 1891-1904.
- Pomeroy J.W., Fang X. and Marks D.G. (2016): The cold rain-on-snow event of June
in the Canadian Rockies characteristics and diagnosis. Hydrological Processes, 30,
  2899-2914. DOI: 10.1002/hyp.10905
- Schirmer, M., & Lehning, M. (2011). Persistence in intra-annual snow depth
  distribution: 2. Fractal analysis of snow depth development. Water Resources Research,
  W09516.
- Schirmer, M., Wirz, V., Clifton, A., & Lehning, M. (2011). Persistence in intra-annual
  snow depth distribution: 1. Measurements and topographic control. Water Resources
  Research, W09517.
- Shook, K. & Gray, D. M. (1996). Small-scale spatial structure of shallow snowcovers.
  Hydrological Processes, 10, 1283-1292.
- Skiles, S. M., Painter, T. H., Belnap, J., Holland, L., Reynolds, R. L., Goldstein, H. L.
  & Lin, J. (2015). Regional variability in dust-on-snow processes and impacts in the Upper
  Colorado River Basin. Hydrological Processes, 29, 5397-5413.
- Trujillo, E., Ramírez, J. A., & Elder, K. J. (2007). Topographic, meteorologic, and
  canopy controls on the scaling characteristics of the spatial distribution of snow depth fields.
  Water Resources Research, 43(7).

| 1  | Wayand, N. E., Marsh, C. B., Shea, J. M., & Pomeroy, J. W. (Verseghy, D. L. (2000).         |
|----|---------------------------------------------------------------------------------------------|
| 2  | The Canadian land surface scheme (CLASS): Its history and future. Atmosphere Ocean, 38,     |
| 3  | <del>1-13.</del>                                                                            |
| 4  | 2018). Globally scalable alpine snow metrics. Remote Sensing of Environment, 213,           |
| 5  | 61-72.                                                                               |
| 6  | Webster, R. and Oliver, M. (2007): Geostatistics for Environmental Scientists.              |
| 7  | Westoby, M., Brasington, J., Glasser, N., Hambrey, M. & Reynolds, J. (2012):                |
| 8  | "Structure-from-Motion" photogrammetry: A low-cost, effective tool for geoscience           |
| 9  | applications, Geomorphology, 179, 300-314.                                                  |
| 10 | Winstral, A., & Marks, D. (2002). Simulating wind fields and snow redistribution            |
| 11 | using terrain-based parameters to model snow accumulation and melt over a semi-arid         |
| 12 | mountain catchment. Hydrological Processes, 16(18), 3585-3603.                              |
| 13 | Winstral, A., Marks, D., & Gurney, R. (2013). Simulating wind affected snow                 |
| 14 | accumulations at catchment to basin scales. Advances in Water Resources, 55, 64-79.         |
| 15 | Winstral, A., & Marks, D. (2014). Long-term snow distribution observations in a             |
| 16 | mountain catchment: Assessing variability, time stability, and the representativeness of an |
| 17 | index site. Water Resources Research, 50(1), 293-305.                                       |

Figure 1. Extraterrestrial solar irradiance at 50° N for north, south and east/west facing 30 % slopes. Note the small differences as summer solstice is approached (<del>Maleafter Gray and <del>Gray</del>Male, 1981).</del>

---

## Author Response (AR2)

**Reply to Charles Luce**

We thank Charles Luce for the constructive comments. We copied the comments for a better readability and marked them using italic fonts.

*The authors have done a nice job revising the paper. The introduction is much more succinct and focused on the problem needing to be solved, and the novel contribution is outlined and given in the conclusions.*

*There is a little bit of new writing that left some concern for me. P4 lines 20-25 presents what seems to be not entirely a fair, nor necessarily accurate, criticism of using empirical modeling of initial SWE variability. While the Dornes citation is given for inconsistency between years (but also discusses correlations with melt, which steps outside of the initial SWE variability issue), others, Luce and Tarboton (2004) as one example, show the opposite, relatively consistent SCD curves across years (at least when presented dimensionlessly, which is not how everyone describes SCD curves). The Dornes work certainly gives pause with respect to generality, but this paragraph gives the impression of concluding that one should not use initial SWE variability based on empirical information period. At the same time, the current manuscript eventually concludes that initial SWE variability is the most important factor in the current study, but does not actually assess interannual variability in the initial distribution. The concluding sentence states that it is not possible to evaluate the impacts of correlations without first knowing the contribution from initial variability. Again, the critiqued methods do just that (at least one of them does). It seems strange to criticize these methods for this issue in this manuscript, when this manuscript mostly notes that scale mismatches may be a key reason for not finding spatial correlations between melt and initial SWE distributions. If this paragraph fits into this paper, a more balanced discussion is necessary. It seems reasonable to see the issue presented as a question, or even a point of difference among sites that could be measured with replication, but the current way it is presented makes the issue look settled.*

We agree with Charles Luce that this citation which is showing inconsistencies between years, is not a good way to move over to the topic of our study, which is showing only observational data for one year in detail. Thus, we extended a previous citation of Dornes work some lines earlier in a more balanced way and added a model approach using high-resolution LiDAR data to generate SWE variability (Brauchli et al., 2017).

**Reply to the anonymous reviewer**

We thank the reviewer for the constructive comments. We copied the comments for a better readability and marked them using italic fonts.

*The authors have addressed all of my comments and improved the readability of the manuscript. I have just a few minor comments that they might want to consider before resubmission. All comments refer to the most recent version (without track changes).*

*P1L16: I would suggest to introduce "HS" as abbreviation for "snow depth" here and to use it throughout the abstract.*

The used abbreviation dHS stands for 'difference in snow depth'. We also use the abbreviation HS for 'snow depth'.

*P1L28: "Analysing the reasons for a missing correlation in this study area provided some prerequisites for large spatial correlations and for when these need to be taken into account by SCD curves." What do you mean by "some prerequisites for large spatial correlations"?*

We reformulated this sentence.

*P2L1-2: "models […] must account for correlations between SWE and melt […]", this seems to contradict your earlier statement that there is "… only a weak spatial correlation developed between initial snow depth and dHS".*

This is indeed misunderstanding, since we meant that incorporating a wrong spatial correlation in a model is a large modelling mistake. Wrong can be large in the model and small in the real world, or the other way round. We reformulated this sentence.

*P2L32: Missing "that" in "There are studies have found…"*

Done as suggested.

*P2L10: Remove "by" in "LiDAR by…"*

Done as suggested.

*P3L18-22: "On the other side, spatially varying melt rates […] can alter this pre-melt SWE distribution and, when correlated to SWE, result in a spatial variability to SCD […]" This statement is confusing. Do you mean that pre-melt SWE is correlated to SWE which is measured during melt? What is spatially variable to SCD? Is this a grammar error?*

We reformulated this sentence.

*P7L2-3: "As such HS and dHS are analysed and interpreted as proxies for SWE and melt in the text." I*

*would have thought that dHS is a proxy for ablation (mainly melt and redistribution) rather than melt alone.*

Redistribution of snow is not relevant in this late ablation season anymore. We included this aspect in the sentence.

*P13L1-2: "The lack of correlation in the Fig. 5c point cloud was contributed to by compensation between the positive correlation driven by melt energy and the negative correlation from a cold content." I do not understand from the previous discussion of Figure 5c (P12L17-31) how you can reach this conclusion. Melt energy and cold content were not explicitly analyzed, so your statement seems to be rather hypothetical. Although you discuss melt energy and cold content much later on P17, it might be useful to add a reference to this discussion to the earlier statement on P12.*

We made the hypothetical character of this statement more clear.

*P13L14-15: "Remote sensing techniques such as remote sensing can determine where deep drifts occur on north-facing slopes […]"*

Changed.

*P14L6-14: How was this correlation analysis done? Did you compare spatial or temporal patterns of ablation (dHS)?*

This is a temporal correlation of spatial patterns. Spatial patterns were analyzed pixel-by-pixel. We added this information.

*P17L11-12: "The large scatter between HS0 and dHS results from the observation that most of the variance of HS0 occurs at much smaller scales (Fig. 7a)." Doesn't this statement refer to Figure 8c? Figure 7a refers to dHS, not HS0.*

Changed. Many thanks for finding this error.

*Figures 2-6: Sub-plot references (a, b, c and d) are missing in all figures.*

Sub-plot references will be inserted in each image files, which will be finally submitted.

*Figure 5: What are the red, blue and yellow data points in Figure 5c? Please indicate in the caption that you show data of P1 only.*

Done as suggested.

*Figure 8: The caption seems incomplete, e.g., what are "dHS.irradiance" and "dHS.HS0"?*

Changed.

[revised manuscript text omitted]